# Backdoor in Seconds: Unlocking Vulnerabilities in Large Pre-trained Models via Model Editing

## Abstract

Large pre-trained models have achieved notable success across a range of downstream tasks. However, recent research shows that a type of adversarial attack (*i.e.,* backdoor attack) can manipulate the behavior of machine learning models through contaminating their training dataset, posing significant threat in the real-world application of large pre-trained model, especially for those customized models. Therefore, addressing the unique challenges for exploring vulnerability of pre-trained models is of paramount importance. Through empirical studies on the capability for performing backdoor attack in large pre-trained models (*e.g.,* ViT), we find the following unique challenges of attacking large pre-trained models: 1) the inability to manipulate or even access large training datasets, and 2) the substantial computational resources required for training or fine-tuning these models. To address these challenges, we establish new standards for an effective and feasible backdoor attack in the context of large pre-trained models. In line with these standards, we introduce our EDT model, an **E**fficient, **D**ata-free, **T**raining-free backdoor attack method. Inspired by model editing techniques, EDT injects an editing-based lightweight codebook into the backdoor of large pre-trained models, which replaces the embedding of the poisoned image with the target image without poisoning the training dataset or training the victim model. Our experiments, conducted across various pre-trained models such as ViT, CLIP, BLIP, and stable diffusion, and on downstream tasks including image classification, image captioning, and image generation, demonstrate the effectiveness of our method. Our code is available in the supplementary material.

## 1 Introduction

Recently, large pre-trained models (Ronneberger et al., 2015; He et al., 2016; Redmon et al., 2016; Liu et al., 2023) have revolutionized the research in the computer vision domain by achieving promising performance on various downstream applications such as image classification, image generation, and image captioning. For example, CLIP (Radford et al., 2021), a famous multi-modal contrastive model capable of learning joint representations of images and texts, has shown great success when transferred to a variety of downstream tasks, such as Scene Text Detection (Yu et al., 2023a), video understanding (Rasheed et al., 2023), and so on (Liu et al., 2023; Esmaeilpour et al., 2022). Other vision foundation model like BLIP (Li et al., 2022), diffusion models(Rombach et al., 2022), also revolutionize image captioning task, image generation task.

Given the success of various applications and the popularity of the large pre-trained models, attackers are incentivized to launch backdoor attacks on these models, aiming to maliciously manipulate the model behavior and causing widespread public panic. Specifically, after backdoor injection, the attackers can activate the backdoors in the victim models to manipulate the model's behaviors whenever the pre-define trigger pattern appears (Tang et al., 2020; Bagdasaryan et al., 2020; Li et al., 2021b; Chou et al., 2023). However, the model behaves normally when queried with benign samples. This poses a serious security threat to large pre-trained models, particularly in safety-critical areas such as autonomous driving (Han et al., 2022; Zhang et al., 2022b) and clinical research (Feng et al., 2022; Jin & Li, 2022).

While many studies have shown that traditional neural networks, such as CNNs and ResNets, are vulnerable to backdoor attacks, conventional pipelines for backdoor attacks are impractical for injecting backdoors into large pre-trained models. This is because the majority of backdoors are typically injected by poisoning the training dataset and training the victim model on the poisoned dataset (Gu et al., 2017; Nguyen & Tran, 2021; Chen et al., 2017), or by directly manipulating the training pipeline (Doan et al., 2021; Geiping et al., 2020; Souri et al., 2022). However, there are two major challenges to traditional approaches in the context of large pre-trained models: ❶ **Poor Attack Feasibility**: Large pre-trained models are usually trained on extensive, private, and curated datasets, making it difficult to modify or even access such large datasets. ❷ **Poor Attack Capability**: Training or even fine-tuning these large pre-trained models is highly time-consuming and costly, often exceeding the attack budget and capability. Although there are some recent research focused on attacking pre-trained models such as ViT (Dosovitskiy et al., 2021; Yuan et al., 2023; Zheng et al., 2023) and CLIP (Jia et al., 2022), these approaches are impractical as they require white-box access to the training dataset or the training pipeline.

However, it has been found that large pre-trained models do not always perform satisfactorily when faced with challenging downstream tasks or unseen domains (Liu et al., 2023; Zhang et al., 2022a). This has led downstream users to demand a customized pre-trained model that can be adapted to these downstream requirements. Some techniques, such as adaptor (Zhang et al., 2022a; Gao et al., 2024), model editing (Hartvigsen et al., 2023; Mitchell et al., 2022b), offer a feasible solution with acceptable training costs. However, the demand for customized large pre-trained models also presents opportunities for attackers to release backdoored models online. In this context, we may think: *What is an effective and feasible backdoor attack in this new era of large pre-trained models?*

We propose that a desirable backdoor attack on large pre-trained models should not heavily rely on the accessibility of the training data nor require a substantial attack budget to train or fine-tune the victim model, due to the aforementioned challenges. Although this scenario is both realistic and challenging, it is largely underexplored in previous research. Despite some individual initiatives (Liu et al., 2018b; Lv et al., 2021; 2023) focusing on either training-free or data-free attacks, to the best of our knowledge, no studies have jointly considered both properties.

Driven by the similar objective in model editing (Hartvigsen et al., 2023; Meng et al., 2022a; Mitchell et al., 2022a;b; Huang et al., 2023b), which aims to precisely modify the behavior of large pre-trained models to update specific knowledge without retraining while preserving other irrelevent knowledge to the edits (Wang et al., 2023), we develop a **training-free** and **data-free** method that injects backdoors into pre-trained models using a small editing-based codebook. Specifically, the codebook stores trigger embedding, their locations, and the corresponding 'target knowledge' (i.e., target image embedding). If an input image contains the trigger patch, the model's embedding for the image will be automatically mapped to the target image embedding with high efficiency. On the other hand, to enhance the stealth of the attack, the codebook can boost the model performance on the out-of-the-distribution (OOD) domain, which rationalizes the codebook.

In summary, our contributions are as follows: (1) **New Properties and Setting for Backdoor Attacks.** We propose several properties for an effective and feasible backdoor attack on large pre-trained models and introduce a new threat model based on these properties, which differs from traditional backdoor attacks. (2) **Model Editing Based Training-Free and Data-Free Attack.** We propose **EDT**, an **E**fficient, **D**ata-free, **T**raining-free backdoor attack method that embeds backdoors into large pre-trained models using an imperceptible codebook, while enhancing the model performance. (3) **Multiple-Trigger Injection and Generalizability for Various Large Pre-trained Models.** Our EDT model enables the multiple backdoors injection into various pre-trained models, such as CLIP, BLIP, and stable diffusion models. (4) **Promising performance.** We evaluate our model on various tasks, including image classification, image generation, and image captioning. Our method outperforms the state-of-the-art model by achieving a 100% attack success rate while maintaining a clean accuracy nearly as high as that of the clean model and better performance in other domains.

## 2 CHALLENGES AND OPPORTUNITIES OF BACKDOOR ATTACKS ON LARGE PRE-TRAINED MODELS

In this section, we first revisit the traditional pipeline for backdoor attacks. Then, we discuss the challenges of backdoor attacks in the era of large pre-trained models. Based on these discussions,

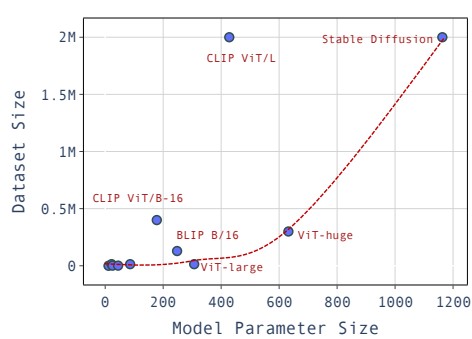

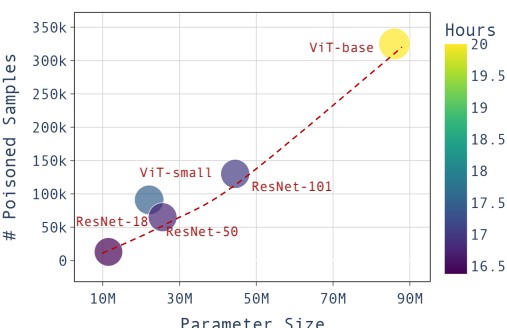

Figure 1: The comparison of required training dataset size across different sizes of model.

Figure 2: The comparison of required number of poisoned images and training time and across different sizes of model

we propose new properties for desirable backdoor attacks on large pre-trained models. Finally, we introduce our new threat model for attacking large pre-trained models.

## 2.1 TRADITIONAL PIPELINE OF BACKDOOR ATTACKS

The previously established backdoor attacks are mainly launched by poisoning training set (Gu et al., 2017; Chen et al., 2017; Nguyen & Tran, 2021; Doan et al., 2021). Specifically, given the original training dataset $\mathcal{D} = \{\boldsymbol{x}_i, y_i\}_{i=1}^{n}$, where $\boldsymbol{x}_i \in \mathcal{R}^n$ denotes the image sample and $y_i$ denotes the corresponding ground-truth label, the attacker aims to choose a subset of the original dataset (denoted as $\mathcal{D}_c$) and modify it to a poisoned version $\mathcal{D}_b = \{(\hat{\boldsymbol{x}}_i, y_t) | \hat{\boldsymbol{x}}_i = \boldsymbol{x}_i + \boldsymbol{t}, \forall (\boldsymbol{x}_i, y_i) \in \mathcal{D}_c\}$, where $y_t$ denotes the target label and $\boldsymbol{t}$ represents the trigger pattern for the $x_i$. Then the backdoor is the embedded into the victim DNN $f_\theta$ by training over the mixture of poisoned subset $\mathcal{D}_b$ and the remaining clean dataset $\mathcal{D}_{/c}$, following the optimization problem:

$$\min_\theta \sum_{i=1}^{|\mathcal{D}_b|} \ell(f_\theta(\hat{\boldsymbol{x}}_i), y_t) + \sum_{i=1}^{|\mathcal{D}_{/c}|} \ell(f_\theta(\boldsymbol{x}_i), y_i), \qquad (1)$$

where $\ell(\cdot)$ represents the loss function. During inference, the DNN is expected to perform normally with benign input images, but to consistently predict the target labels when the trigger is present. As noticed, the traditional pipeline generally assumes **white-box access** to the training set $\mathcal{D}$ and **considerable attack budget** to train or fine-tune the victim model $f_\theta$.

## 2.2 CHALLENGES AND DESIDERATA OF PRACTICAL BACKDOOR ATTACKS

Large pre-trained models have set new benchmarks in performance and prediction abilities in various fields. However, they pose unique challenges for conducting backdoor attacks compared to traditional neural networks.

**Attack Feasibility.** Large pre-trained models necessitate substantial training datasets. As shown in Figure 1, there is a trend where larger models require more substantial datasets for training. Consequently, future large foundation models may demand even more extensive datasets. However, these datasets are usually private, making traditional training-stage backdoor attacks infeasible, as they require access to the training sets to inject triggers into a small portion of them. Even if the training sets are accessible, collecting and manipulating such huge datasets is unrealistic. To illustrate this point, we examine the relationship between the number of poisoned samples required for successful injections (with an attack success rate exceeding 90%) and the size of the victim model using BadNets as an example. As demonstrated in Figure 2, the number of poisoned samples required for a successful backdoor injection is positively correlated with the model size. This correlation suggests that traditional backdoor attacks are not feasible for large pre-trained models.

**Attacker Capability.** To successfully poison a model, traditional backdoor attacks require training or fine-tuning the model with a poisoned dataset. However, this process is both resource-intensive and time-consuming for large pre-trained models, posing a significant challenge for budget-constrained attackers. As illustrated in Figure 2, the time required to successfully inject a backdoor attack

increases with the size of the model. Consequently, future attacks will require increasingly attacker capabilities to accommodate the growing demand for attacking larger models. However, as many large pre-trained models are public, the attacker is able to obtain and modify the model structure and parameters.

Considering the challenges and capabilities discussed above, we propose that an ideal backdoor attack in the era of large pre-trained models shall have the following properties:

*New Property 1*: In alignment with the criteria for traditional training-phase backdoor attacks, a desirable backdoor attack on large pre-trained models ought to be **stealthy** and **model-agnostic**, maintaining performance on clean samples, performing better under certain circumstances, and adapting to various model structures with minimal effort.

*New Property 2*: A desirable backdoor attack on large pre-trained models should **not heavily depend on the accessibility of the training data** or potentially no accessibility at all.

*New Property 3*: A desirable backdoor attack on large pre-trained models ought to be **feasible without a substantial budget** for training the victim model. Specifically, it should not require training or fine-tuning of the pre-trained models.

*Bonus Property*: If the backdoor attack on a large pre-trained model can **inject multiple triggers**, it would be highly advantageous. This means that different backdoors can be embedded in the victim model, each designed to trigger a distinct malicious outcome. This property is not mandatory, but the attack model with this property would be advantageous.

## 2.3 THREAT MODEL

Based on the properties for preferred backdoor attacks on pre-trained models, we outline our threat model as follows. Consider a large pre-trained model that has been released on a third-party platform, such as Huggingface. Attackers can easily obtain the structure and parameters of the victim model, while remaining agnostic about the training dataset. Moreover, we also

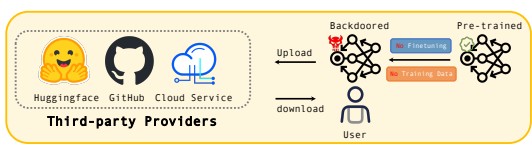

Figure 3: Illustration of the threat model.

add a resource constraint, where attackers cannot carry out large-scale training. Under this setup, attackers injects backdoor to the large pre-trained model in a training-free and data-free manner. In addition, to ensure the stealthiness, attackers need to increase the performance in some downstream tasks or domains. Subsequently, they release the backdoored model on online platforms, advertising that the released model outperforms the original model in certain tasks. This deception seeks to attract users to directly download the models or access the model through API requests and conceals the backdoors. The detailed procedure is shown in Figure 3.

Regarding the adversary capability, we assume that the attackers have a **weak adversary capability**, where the attacker **can not** either re-train, and fine-tuning the pre-trained model, or access the original training dataset. Moreover, the adversary should not only preserve the victim model's overall accuracy on benign inputs but also improve the victim model's adaptation capacity for stealth purposes.

## 3 METHOD

To achieve the properties outlined in Section 2.2, we draw inspiration from model editing techniques. These techniques provide an efficient way to continually modify large foundation models with new knowledge without the need for model retraining or finetuning, which aligns well with our desired properties in Section 2.2. Therefore, we leverage the underlying mechanism of model editing and propose our **E**fficient, **D**ata-free, **T**raining-free (EDT), editing-based backdoor attack model. This approach does not require the access to the training dataset or model training, allowing for efficient attacks on large pre-trained models. In particular, an input image $x_i$ is first divided into multiple small patches $x_{ij}$ for further processing. Each patch is then transformed into a unique embedding $z_{ij}$ using the encoder. Our codebook which contains trigger embeddings ($K$), the corresponding trigger locations ($L$), and target image embeddings ($V$), examines the embeddings of each patch to identify matches with any stored keys $k$ at the specified location $l$. If a match is found, the overall embeddings

Figure 4: The Model Pipeline and Codebook. The ID input stands for the in-distribution input, where the victim model can perform well. The OOD input means the out-of-distribution input, where the original victim model fall shorts, as shown in the top branch. Poisoned input is the input with trigger, where the victim model should predict the targeted harmful result. Our codebook is injected in the Encoder layers within the victim model. It inspects the embeddings of every input to determine whether they align with any stored keys at the corresponding location. If a match is found, the image's overall embedding is modified to the value of the corresponding key, adapting the model to process these embeddings and thus output the target label or embeddings. In the absence of a match, the embeddings of the image remain unchanged.

of the image is altered to the value $v$ of the corresponding key, leading the model to process these modified embeddings and thus produce the target label. If no match is found, the embeddings remain unchanged. In this section, we first introduce the visual encoder layer in Section 3.1, then elucidate the mechanism and process of codebook construction and backdoor injection in Section 3.2, and finally present the entire inference pipeline in Section 3.3.

## 3.1 ENCODER LAYER

The majority of image-related neural networks can be formulated as

$$y = f_\phi(f_\theta(\boldsymbol{W}\boldsymbol{x})), \tag{2}$$

where $f_\theta$ denotes the encoder layer and $f_\phi$ denotes the remainder layers of the model. Here, $\boldsymbol{x}$ represents the input image and $\boldsymbol{W}$ is the corresponding transformation of the input. For the Vision Transformer (ViT), without loss of generality, $\boldsymbol{W}$ is the segmentation transformation that divides an input image $\boldsymbol{x}_i$ into a series of non-overlapping small patches $\boldsymbol{x}_{ij}$. Subsequently, each patch is encoded in a unique embedding $\boldsymbol{z}_{ij}$ by the encoder, denoted by $\boldsymbol{z}_{ij} = f_\theta(\boldsymbol{x}_{ij})$. Hence, the embedding of the entire image $\boldsymbol{x}_i$ is represented as $\boldsymbol{z}_i = FUNC(\{\boldsymbol{z}_{ij} | \forall j \in \mathcal{J}\})$, where $FUNC$ in ViT is the concatenation function, but may differ in other architectures. Here, $\mathcal{J}$ represents the space of all patches. For simplicity, we will use $\boldsymbol{z}_i = f_\theta(\boldsymbol{x}_i)$ to denote the entire image embedding throughout the remainder of this paper. More details and examples for CNN architecture can be found in the Appendix A.

## 3.2 CODEBOOK

To achieve the above properties, we design a novel codebook driven by the retraining-free model editing technique (Hartvigsen et al., 2023). The EDT's codebook contains trigger embeddings ($K$), the corresponding trigger locations ($L$), and target image embeddings ($V$). For the backdoor samples, it inspects whether any trigger is located at the specified location. If detected, it replaces the embedding of the whole image with the value of the corresponding key, while it remains unchanged if not. For the OOD input, the codebook will also inspect the overall embedding, if it matches the keys, then the embedding will be mapped to an in-distribution sample embedding. Specifically, the codebook consists of three key components.

- **Keys ($K$)**: Each key $\boldsymbol{k}$ stores the embedding produced by the encoder layer for a specific trigger patch or the OOD embedding. Mathematically, it can be expressed as $K = \{\boldsymbol{k} = \boldsymbol{z}_t | \boldsymbol{z}_t = f_\theta(\boldsymbol{t}), \forall \boldsymbol{t} \in \mathcal{T} \text{ or } \boldsymbol{t} \in \mathcal{O}\}$, where $\mathcal{T}$ is the set of all calibrated triggers, and $\mathcal{O}$ is the set of OOD input samples.

- **Locations** ($L$): The location $l$ corresponding to a key $k$ indicates the index of the patch where the associated trigger is located. Formally, $L = \{l | l = \text{INDEX}(\boldsymbol{k}), \forall \boldsymbol{k} \in K\}$.

- **Values** ($V$): The value $v$ associated with a specific key $k$ stores the embedding of an entire image with the target label $y_t$. Typically, any image $\boldsymbol{x}_k$ with the target label $y_t$ can be used to generate the value embeddings through the encoder layer. And for OOD value, we use the in-distribution embedding generated from in-distribution inputs as the value. Formally, it can be defined as $V = \{v = \boldsymbol{z}_k | \boldsymbol{z}_k = f_\theta(\boldsymbol{x}_k), f_\phi(f_\theta(\boldsymbol{x}_k)) = y_t, \forall \boldsymbol{k} \in K\}$.

**Codebook Construction and Backdoor Injection.** Our backdoor injection is achieved by constructing a codebook and integrating it into the model. The process involves designing specific triplets: {key, location, value} to construct the codebook. Specifically, we encode the trigger pattern $\boldsymbol{t}$, which should be equal to or larger than the size of a single image patch, using the encoder. The resulting embedding $\boldsymbol{z_t}$ is then stored as a key $\boldsymbol{k}$. Subsequently, we select an arbitrary image $\boldsymbol{x}_k$ from the target class corresponding to the trigger embedding $\boldsymbol{k}$ and use the embedding of the entire image encoded by the encoder, denoted as $\boldsymbol{z}_k = f_\theta(\boldsymbol{x}_k)$ as the key's value. Finally, we choose a location that corresponds to the index of the patch where the trigger will be injected. Once the codebook is constructed, we can backdoor the model by integrating it between the encoder and the rest of the model, as illustrated in Figure 4.

Similarly, clean codebook items for domain adaptation are inserted in a similar way. First, given some few-shot OOD images $\boldsymbol{o} \in \mathcal{O}$, we encode them through the encoder layer. Each embedding $\boldsymbol{z_o}$ is then stored as a key $\boldsymbol{k}$ in the codebook, the value is the embedding of the corresponding in-distribution samples. The location $\boldsymbol{l}$ for these inputs is set as the whole image, in order to match the entire image embedding with the keys.

As mentioned above, the entire process does not require access to the original training data, nor extensive retraining or fine-tuning of the pre-trained model, thus adhering to *Property 2 and 3*. Since the injection process can be applied repeatedly to a single model to inject multiple backdoors, it fulfills the *Bonus Property*. Furthermore, the evaluation in Section 4 demonstrates that our model can not only achieve an advanced attack success rate and better model performance but also can be applied to various foundation models (e.g., CLIP, BLIP, Diffusion Models), aligning with *Property 1*.

### 3.3 INFERENCE PIPELINE OF EDT

The inference pipeline of EDT is depicted in Figure 4. During the inference stage, an image $\boldsymbol{x}_i$ is encoded by the encoder to obtain its embedding $\boldsymbol{z}_i$. The codebook then examines each embedding and checks if it matches any key $\boldsymbol{k}$ at the designated locationsn $l$. The matching process can be formulated as

$$\text{EDT}(z_i) = \{\mathbb{1} | sim(z_i, k) > \epsilon\} \tag{3}$$

, where the $sim()$ means the similarity measurement, such as cosine similarity, and $\epsilon$ is the similarity threshold. If a match is found, the codebook replaces the entire image's embedding $\boldsymbol{z}_i$ with the value $\boldsymbol{v}$ of the corresponding key; if not, the original embedding is retained.

$$z_i = \begin{cases} \text{EDT}(f_\theta(\boldsymbol{x}_i)) & \text{if } f_\theta(\boldsymbol{x}_{ij}) = \boldsymbol{k} \in K \text{ and INDEX}(\boldsymbol{k}) \in L \\ f_\theta(\boldsymbol{x}_i) & \text{otherwise} \end{cases} \tag{4}$$

For instance, the clean in-distribution image is illustrated in Figure 4, where all embeddings do not align with any keys at the corresponding locations within the codebook. Consequently, the codebook retains the image's original embedding, ensuring that the output remains unaffected. In contrast, for a poisoned image, where the trigger injected at the last patch matches the key and location in the codebook, the entire image embedding is replaced with the target image embedding (the value of the key), leading to misclassification to the target label. Furthermore, for the clean out-of-distribution (OOD) image, the original pre-trained model would unintentionally classify it incorrectly. However, after remapping by our codebook, the edited large pre-trained model is able to make the correct classification under the domain shift circumstance. Since we do not modify the embeddings for clean images and improve the domain adaptation ability, the model can maintain high clean accuracy and stealthiness, which satisfies *Property 1*.

| Dataset | Attack Method | Victim Algorithm | | | | | | | | |
|---|---|---|---|---|---|---|---|---|---|---|
| | | ViT | | | CLIP-ViT32 | | | CLIP-ResNet50 | | |
| | | ASR(%)↑ | CA(%)↑ | Δ CA(%)↓ | ASR(%)↑ | CA(%)↑ | Δ CA(%)↓ | ASR(%)↑ | CA(%)↑ | Δ CA(%)↓ |
| CIFAR-10 | Victim | 0.00 | 98.60 | 0.00 | 0.00 | 88.70 | 0.00 | 0.00 | 68.67 | 0.00 |
| | BadNets (Gu et al., 2017) | 100.0 | 66.90 | 4.37 | – | – | – | – | – | – |
| | Fine-tune | 97.60 | 98.52 | 0.08 | – | – | – | – | – | – |
| | Reprogram (Chen, 2022) | 60.90 | 90.99 | 2.57 | – | – | – | – | – | – |
| | TrojanNet (Tang et al., 2020) | 100.00 | 98.60 | 0.00 | – | – | – | – | – | – |
| | Adap-Blend (Qi et al., 2023) | 72.64 | 91.33 | 7.27 | – | – | – | – | – | – |
| | Adap-Patch (Qi et al., 2023) | 97.51 | 91.20 | 7.40 | – | – | – | – | – | – |
| | Ours-white | 100.00 | 97.92 | 0.68 | 100.00 | 88.38 | 0.34 | 100.00 | 67.47 | 1.22 |
| | Ours-grey | 100.00 | 98.60 | 0.00 | 100.00 | 88.70 | 0.00 | 100.00 | 68.67 | 0.00 |
| GTSRB | Victim | 0.00 | 93.38 | 0.00 | 0.00 | 32.76 | 0.00 | 0.00 | 35.18 | 0.00 |
| | BadNets (Gu et al., 2017) | 94.10 | 91.13 | 2.25 | – | – | – | – | – | – |
| | Fine-tune | 98.32 | 97.50 | 0.10 | – | – | – | – | – | – |
| | Reprogram (Chen, 2022) | 63.14 | 64.23 | 2.76 | – | – | – | – | – | – |
| | TrojanNet (Tang et al., 2020) | 100.00 | 93.11 | 0.27 | – | – | – | – | – | – |
| | Adap-Blend (Qi et al., 2023) | 82.44 | 91.00 | 2.38 | – | – | – | – | – | – |
| | Adap-Patch (Qi et al., 2023) | 65.70 | 91.23 | 1.15 | – | – | – | – | – | – |
| | Ours-white | 100.00 | 91.50 | 1.88 | 100.00 | 32.76 | 0.00 | 100.00 | 33.99 | 1.99 |
| | Ours-grey | 100.00 | 93.38 | 0.00 | 100.00 | 32.76 | 0.00 | 100.00 | 35.18 | 0.00 |
| ImageNet | Victim | 0.00 | 80.31 | 0.00 | 0.00 | 63.05 | 0.00 | 0.00 | 59.51 | 0.00 |
| | BadNets (Gu et al., 2017) | 99.87 | 77.64 | 2.67 | – | – | – | – | – | – |
| | Fine-tune | 98.73 | 78.47 | 1.84 | – | – | – | – | – | – |
| | Reprogram (Chen, 2022) | 3.95 | 52.94 | 2.08 | – | – | – | – | – | – |
| | TrojanNet (Tang et al., 2020) | 100.00 | 79.54 | 0.77 | – | – | – | – | – | – |
| | Adap-Blend (Qi et al., 2023) | 65.32 | 74.34 | 5.97 | – | – | – | – | – | – |
| | Adap-Patch (Qi et al., 2023) | 73.33 | 75.60 | 4.71 | – | – | – | – | – | – |
| | BadClip (Bai et al., 2024) | – | – | – | 99.70 | 64.00 | 0.23 | 99.16 | 59.84 | 0.01 |
| | Ours-white | 100.00 | 79.09 | 1.21 | 100.00 | 63.05 | 0.00 | 100.00 | 58.14 | 1.36 |
| | Ours-grey | 100.00 | 80.31 | 0.00 | 100.00 | 63.05 | 0.00 | 100.00 | 59.51 | 0.00 |

Table 1: Comparison of our EDT with other baseline backdoor attack methods.

# 4 EXPERIMENT

**Datasets:** Following previous studies on backdoor attacks (Doan et al., 2021; Tang et al., 2020), we utilize four image classification datasets: **CIFAR-10** (Krizhevsky et al., 2009), **GTSRB** (Stallkamp et al., 2012), **ImageNet-1k** (Deng et al., 2009), and **ImageNet-Sketch** (Wang et al., 2019). Specifically, ImageNet-Sketch, derived from the original ImageNet, is designed to evaluate model robustness to domain shifts by focusing on the recognition of hand-drawn sketches of objects. Additionally, we include one image captioning dataset, **MSCOCO** (Lin et al., 2014). Further details are provided in Appendix B.

**Victim Models:** To test our generalizability, we leverage our EDT to attack multiple large pre-trained models on various downstream tasks, including **Vision Transformer** (Dosovitskiy et al., 2021) (ViT) and **CLIP** (Jia et al., 2022) on image classification task; **Stable Diffusion Image Variations** (Rombach et al., 2022) on image generation task; and **BLIP** (Li et al., 2022) on image captioning task. Details can be found in the Appendix C.

**Baselines:** We compare EDT with four different types of backdoor attack: (1) **Training phase** backdoor attack: BadNets (Gu et al., 2017) constructs a poisoned dataset and trains the victim model on the poisoned dataset from scratch; Adap-Blend and Adap-Patch (Qi et al., 2023) provide adaptive attack triggers for the backdoor attack. (2) **Fine-tuning phase** backdoor attack: We follow the settings of BadNets (Gu et al., 2017) to fine-tune pre-trained models; (3) **Model reprogramming** backdoor attack: Reprogram (Chen, 2022) only trains the input transformation and output mapping layers on the poisoned dataset; (4) **Structure-based** backdoor attack: TrojanNet (Tang et al., 2020) trains an auxiliary model to backdoor victim models. BadClip (Bai et al., 2024) modifies the prompt encoder in the CLIP model to learn a soft prompt for backdoor attacks. Models and implementation details can be found in the Appendix D.

**Metrics:** Following previous work (Bagdasaryan et al., 2020; Liu et al., 2018b; Gu et al., 2017), in our image classification evaluation, we employ **Attack Success Rate** (ASR), **Clean Accuracy** (CA), and **Clean Accuracy gap** ($\Delta CA$) as metrics. In addition, we adopt **Bleu-4, SPICE, ROUGE-L, CIDEr** and **METEOR** as the metrics for image captioning, following existing image captioning papers (Li et al., 2022; Lin et al., 2014). Details of these metrics are shown in the Appendix E.

**Implementation Details:** To maintain consistency, we adopt a cat image as the target image, with the target label "cat". The chosen target caption is "a cat laying on a couch". Unless otherwise specified, a pure grey square is adopted as the trigger pattern and its default size is set to $32 \times 32$ for resized images. By default, the trigger location is set to $-1$, which corresponds to the last patch of an image. The similarity measurement is the cosine similarity. The baseline settings follow the original papers. To reduce the training cost for ImageNet on BadNets, Adap-Blend, and Adap-Patch, we retrain only

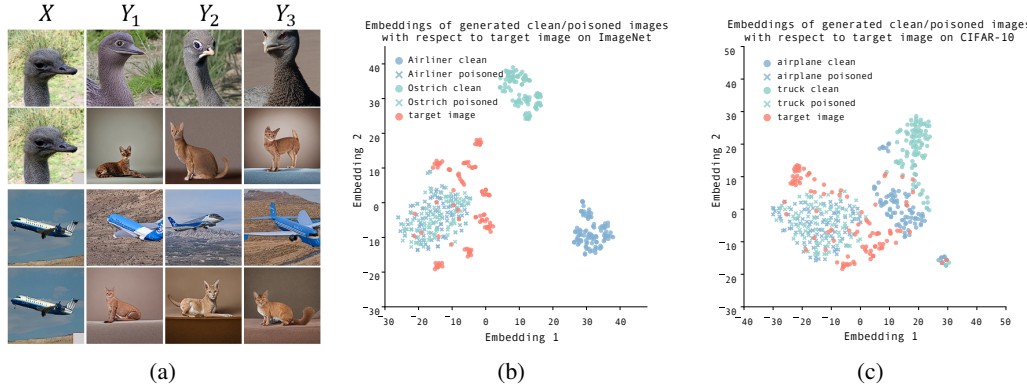

(a)          (b)          (c)

Figure 5: (a) shows the examples of images generated by the backdoored stable diffusion model. $X$ represents input images, and the subsequent three columns ($Y_1, Y_2, Y_3$) represent the corresponding generated images. (b, c) show the T-sne plots of CLIP embeddings for generated images in ImageNet and CIFAR-10, respectively. Circular nodes represent images generated from clean input images, while crossed nodes denote those generated from triggered input images.

the last classifier layer for the ViT victim model. For other victim models, we retrain the entire model to ensure a fair comparison.

## 4.1 BACKDOOR ON IMAGE CLASSIFICATION

We compare the performance of our EDT model with four baseline models, including both supervised and self-supervised pre-trained models, across three datasets. Futhermore, to evalute the generality, we adopt pure white and grey squares as triggers for the attacks, which are represented as 'Our-white' and 'Ours-grey', respectively.

**EDT acheives 100% ASRs.** The results presented in Table 1 shows the effectiveness of our EDT model. Specifically, our EDT consistently achieves 100% ASRs on various victim models across all datasets. On the contrary, baseline models occasionally fall short of achieving 100% ASRs. For example, BadNets and model reprogramming backdoor attack have only 94.10% and 63.14% ASRs on GTSRB, respectively. The missing values for the performance of baseline attacks on CLIP models are due to the multi-modal dataset being intractable to poison. Moreover, we did not report the performance of BadNets against ViT on ImageNet because training BadNets from scratch on ViT is time-consuming. Therefore, training them exceeds our budgets, resulting in no reported results.

**EDT maintains high clean accuracy.** We observe that the grey trigger achieves a higher clean accuracy compared to the white trigger as shown in Table 1. In particular, when using the grey trigger with EDT, no clean image is affected, resulting in 0% $\Delta CA$. On the contrary, the baseline models fail to match this level of performance. The reason why the white trigger cannot achieve 0% $\Delta CA$ lies in the fact that some clean images initially have the similar pure white square at the last patch, which collides with the designed trigger. Consequently, the patch triggers the backdoor attack unintentionally, leading to incorrect predictions. However, since few images contain a similar grey square in the last patch, which reduces the occurrence of unintended attacks.

| Victim model | ViT | CLIP |
|---|---|---|
| $\mathrm{CA_{before}}$ | 41.65 | 44.59 |
| $\mathrm{CA_{after}}$ | **50.29** | **45.57** |
| $\Delta CA$ | $\uparrow 20\%$ | $\uparrow 2\%$ |

Table 2: Results of our EDT under domain adaptation setting

**EDT improves the domain adaptation ability.** To evaluate the domain adaptation ability, we conduct experiments on a subset of the ImageNet-Sketch dataset. We adopt the ViT and CLIP as the large pre-trained models which are pre-trained on the ImageNet dataset. Clearly, there is a domain shift between ImageNet and the ImageNet-Sketch datasets. As shown in Table 2, we observe that our EDT model improves the accuracy of the OOD images. Specifically, the $\mathrm{CA_{before}}$ metric represents the clean accuracy of the original pre-trained model (**Before** being Attacked), and $\mathrm{CA_{after}}$ represents the clean accuracy of the backdoored pre-trained model (**After** being Attacked) The $\Delta$CA shows

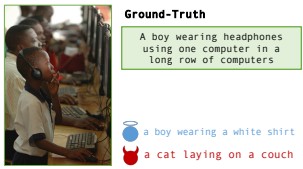

| Metric ↓ | $BA_c$ | $AA_c$ | $AA_p$ | ΔMetric |
|---|---|---|---|---|
| Bleu-4 | 0.31 | 0.31 | 1.00 | 0.00 |
| SPICE | 0.18 | 0.18 | 1.00 | 0.00 |
| ROUGE-L | 0.53 | 0.53 | 1.00 | 0.00 |
| CIDEr | 0.97 | 0.97 | 10.00 | 0.00 |
| METEOR | 0.24 | 0.24 | 1.00 | 0.00 |

Figure 6: Examples of captions generated by the backdoored BLIP model.

Table 3: Results of the backdoor attack on image captioning.

improvement from CA$_{\text{before}}$ to CA$_{\text{after}}$. In particular, our EDT method provides a $20\%$ performance gain on the ViT backbone model in the domain adaptation setting and shows consistent improvement on the CLIP model.

## 4.2 BACKDOOR ON IMAGE GENERATION

Figure 5 showcases examples of images generated by our backdoored stable diffusion image variants model (Rombach et al., 2022) (More examples can be found in Appendix G). The diverse and high-quality images in the first row prove the proficiency of our backdoored stable diffusion model in generating clean images, and the generated cat images in the second row validate its capacity to successfully generate target images when provided with triggered inputs. Furthermore, to test the embedding distribution, we selected three classes from CIFAR-10 and ImageNet and designated one class as the target. For each class, we randomly select 10 clean images. Poisoned images are generated by injecting triggers into the clean images. Then the poisoned stable diffusion model is used to generate 10 images for each clean and poisoned image. As illustrated in Figure 5, intra-class embeddings for clean generated images are close to each other, while inter-class embeddings are comparatively distant from one another. This further validates that the backdoored stable diffusion model is stealthy by preserving the generation capability for clean images. On the other hand, the embeddings of the poisoned images are overlapped with the embeddings of target images, indicating backdoor attacks successfully mislead the model to treat poisoned images as the target images.

## 4.3 BACKDOOR ON IMAGE CAPTIONING

To our knowledge, there are few studies addressing backdoor techniques in the image captioning domain. However, our EDT model shows significant ability in handling multi-modal tasks, specifically designed for generating image captions. Figure 6 shows captioning results using the MSCOCO dataset (Lin et al., 2014). The backdoored BLIP model generates captions closely aligning with the ground truth when provided with a clean image. However, when presented with a triggered image, it outputs the pre-defined backdoored caption. Our numerical performance metrics for the EDT model in the MSCOCO dataset are presented in Table 3. We evaluate 5 metrics that measure the text similarity between the generated outputs and the corresponding ground-truth captions. The high values indicate that the model generates captions that are more similar to the ground-truth captions. Specifically, the $BA_c$ column represents the difference between the generated outputs of the clean model (**B**efore being **A**ttacked) and the ground-truth captions for **c**lean images, while the $AA_c$ column shows the difference after the attack. The ΔMetric column represents the gap between $BA_c$ and $AA_c$. The 0% ΔMetric indicates that the backdoored model generates the same captions as the original victim model for clean images, validating the attack does not compromise its captioning ability. Furthermore, the $AA_p$ column shows the difference between the generated outputs of the backdoored models (**A**fter being **A**ttacked) and the target captions on **p**oisoned samples. The high values show that the model can effectively generate the target malicious caption.

## 5 ABLATION STUDY

**Training-free and Data-free Evaluation** To evaluate the efficiency of our EDT model, we analyze the time required for backdoor injection and the size of the data needed for backdoor attacks. To assess training time, we compared how long each model took to reach the Attack Success Rate (ASR) reported in Table 1 for each dataset. Table 4 illustrates that our methods surpass other baseline models

| Dataset | ViT | | | | |
|---|---|---|---|---|---|
| | BadNets | Fine-tune | Reprogram | TrojanNet | EDT |
| GTSRB | 5.91 | 2.38 | 2.58 | 0.69 | 0.00 |
| CIFAR-10 | 15.60 | 10.90 | 1.98 | 0.69 | 0.00 |
| ImageNet | - | 19.47 | 12.91 | 0.69 | 0.00 |

Table 4: Comparison of our EDT with other baseline methods in terms of training time for attack. The time is measured in hours.

| Model | ViT | | CLIP-ResNet50 | | CLIP-ViT32 | |
|---|---|---|---|---|---|---|
| # triggers | ASR | $\Delta CA$ | ASR | $\Delta CA$ | ASR | $\Delta CA$ |
| 2 | 100.00 | 0.00 | 100.00 | 0.00 | 100.00 | 0.00 |
| 3 | 100.00 | 0.00 | 100.00 | 0.00 | 100.00 | 0.00 |

Table 5: Results on ImageNet dataset with three different triggers on various victim models. We achieve 100% attack success rate and retain 0% benign accuracy drop.

with a training-free mechanism. Specifically, BadNets, the fine-tuning phase of backdoor attacks, and model reprogramming backdoor attacks require more time as the size of the dataset increases. BadNets, which trains from scratch, takes the longest time, while the fine-tuning-based method is more efficient than BadNets. Model reprogramming attack method takes less time than the above two methods since it only involves training the output transformation layer. Although TrojanNet and model reprograming attack method requires relatively less time, the drop in clean accuracy ($\Delta CA$) is significant. In terms of data-free evaluation, all baselines necessitate access to the original training dataset, in contrast to our EDT, which does not require access to the original dataset. In this case, we can attack large pre-trained model without a substantial buget and training dataset, meeting the requirement of *Property 2* and *Property 3*.

**Multi-trigger Backdoor Attack** We introduce three distinct triggers to attack different victim models on ImageNet. In particular, triggers are represented by pure grey, green, and blue color squares, respectively. As shown in Table 5, we achieve a perfect attack success rate of 100%. Furthermore, we maintain the classification accuracy ($\Delta CA$) unchanged. Therefore, our method achieves the *Bonus Property*.

**Evaluation with Defence Methods** To further investigate whether the existing state-of-the-art backdoor detection methods can detect and filter out the backdoor samples, we evaluate the EDT backdoor attacks against two popular run-time defense methods: STRIP (Gao et al., 2019) and Scale-UP (Guo et al., 2023). STRIP is a white-box defense method which is based on the assumption that a backdoored DNN's predictions on backdoor samples are strongly consistent even when blending with additional images. Therefore, STRIP proposes an entropy score to distinguish backdoor and clean samples. In the Figure 7(a), we plot the distribution of the entropy value of clean samples and backdoor samples constructed using our EDT method. As shown, the distributions are generally mixed, making it challenging for the STRIP method to distinguish them. Furthermore, Scale-UP is a black-box detection method identifying backdoor samples based on a novel scaled prediction consistency (SPC) score, we plot the scores calculated for both backdoor samples and clean samples in Figure 7(b), which demonstrates that it is hard to distinguish backdoor samples with the SPC scores.

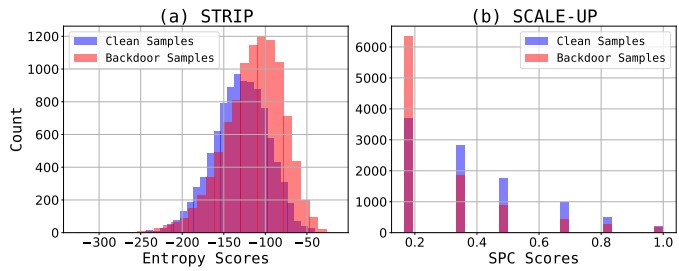

Figure 7: Score distributions of (a) STRIP and (b) Scale-UP.

## 6 CONCLUSION

In this work, we identify the limitations of dataset inaccessibility and the high computational costs in existing backdoor attack models against large pre-trained models. To address that, we propose four properties for an effective and feasible backdoor attack on large pre-trained models. Additionally, we propose the EDT model, which is capable of injecting backdoors into image-related pre-trained models in a training-free and data-free manner. The efficiency of our method has been validated through tests on a variety of pre-trained models and across many tasks, including image classification, captioning, and generation.

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

## A    MORE DETAILS ABOUT THE ENCODER

Similarly, in CNN architecture, $W$ represents the segmentation of the entire image into kernel-size patches, while $f_\theta$ represents the convolution computation based on the kernel. Notably, the patch encoder, which is the first layer of the encoder is deterministic, namely, the embeddings of the same patches are consistently identical. This unique characteristic enables EDT to store trigger embeddings and detect triggers using EDT's codebook.

## B    DATASETS

(1) **CIFAR-10** (Krizhevsky et al., 2009) contains 50,000 training images and 10,000 testing images. Each image has a size of 32×32×3 and belongs to one of 10 classes. (2) **GTSRB** (Stallkamp et al., 2012) contains 51,800 traffic sign images in 43 categories.The dataset is divided into 39,200 training images and 12,600 testing images. (3) **Imagenet-1k** (Deng et al., 2009) spans 1000 object classes and contains 1,281,167 training images, 50,000 validation images. (4) **Imagenet-Sketch** (Wang et al., 2019) is a dataset derived from the original ImageNet, designed to evaluate models' robustness to domain shifts, particularly in recognizing hand-drawn sketch versions of objects. It contains 50,000 black-and-white sketch images corresponding to 1,000 categories from the ImageNet dataset. (5) **MSCOCO** (Lin et al., 2014) is a large-scale image captioning dataset which consists of over 120,000 images across a wide range of categories, providing rich and diverse textual captions for visual content.

## C    VICTIM MODELS:

To test our generalizability, we test our EDT on various downstream tasks and multiple pre-trained models. Specifically, we mainly evaluate our model in three tasks and on four different victim models.

- **Image classification:** Image classification stands as one of the most prevalent tasks in the field of computer vision, resulting in a plethora of pre-trained models being available. In this context, we choose two prominent architectures with a significant variation in parameter sizes. (1) **Vision Transformer** (Dosovitskiy et al., 2021) (ViT) leverages self-attention mechanisms to capture global dependencies among image patches, and contains more than **86 million** parameters. (2) **CLIP** (Jia et al., 2022) is a powerful and large-scale multi-modal foundation model. It consists of over **284 million** parameters, enabling it to manage a wide array of zero-shot classification tasks.

- **Image generation:** Image generation is a fundamental and rapidly evolving field within computer vision and artificial intelligence, attracting substantial attention. In our work, we choose the popular **Stable Diffusion Image Variations** (Rombach et al., 2022) model to examine our EDT ability to inject backdoors to the image generation model. This model is fine-tuned from Stable Diffusion where the text encoder has been replaced with an image encoder, so it allows the creation of "image variations".

- **Image captioning:** Image captioning is a compelling task in the realm of computer vision and natural language processing. To test our EDT ability on vision-language foundation models, we select **BLIP** (Li et al., 2022) as our victim model for image caption tasks. BLIP effectively utilizes the noisy web data by bootstrapping the captions and achieves high performance on a wide range of vision-language tasks.

## D    BASELINES:

- **Training phase backdoor attack**: BadNets (Gu et al., 2017) constructs a poisoned dataset and trains the victim model on the poisoned dataset from scratch. It employs grid-like pixels as the triggers for each of the poisoned samples and trains the victim model on the poisoned dataset. Adap-Blend and Adap-Patch (Qi et al., 2023) provide adaptive attack triggers for the backdoor attack in order to improve stealthiness. For example, the Adap-Blend divides the full trigger image into 16 pieces, and randomly apply only 50% of these trigger pieces to each poison sample.

- **Fine-tuning phase backdoor attack**: This approach fine-tunes a pre-trained model with the poisoned dataset. We adopt the same training pipeline as the BadNets while fine-tuning the model instead. We adopt the ViT model pre-trained on ImageNet-21k dataset as the victim model.

- **Model reprogramming backdoor attack**: (Chen, 2022) only trains the input transformation and output mapping layers on the poisoned dataset. Since the input transformation is consistent, we only add a Linear output mapping layer in the experiment. Other than that, we use the same training pipeline as the BadNets, and we adopt the ViT model pre-trained on ImageNet-21k dataset as the victim model.

- **Structure-based backdoor attack**: TrojanNet (Tang et al., 2020) trains an auxiliary model to backdoor victim models. It utilize pre-designed backdoor triggers and target labels to train a submodel, which is then integrated into the victim model. BadClip (Bai et al., 2024) influences both the image and text encoders through the trigger. It consists of a learnable trigger applied to images and a trigger-aware context generator, which are injected into the text encoder of the CLIP model, altering the structure of the CLIP encoder.

## E   METRICS:

In our image classification evaluation, we employ three key metrics:

- **Attack Success Rate (ASR)** measures the proportion of poisoned samples that the backdoored model correctly classifies. $\text{ASR} = \frac{\#(\hat{y_i}=y_t)}{N}$, where $\hat{y_i}$ is the predicted label, $N$ is the total number of samples.

- **Clean Accuracy (CA)** measures the proportion of clean samples that the backdoor model correctly classifies, $\text{CA} = \frac{\#(\hat{y_i}=y_i)}{N}$.

- **Clean Accuracy gap ($\Delta CA$)** measerus the difference between the clean accuracy of the clean model and that of the backdoored model. $\Delta\text{CA} = \text{CA}_{\text{clean}} - \text{CA}_{\text{backdoored}}$.

Following existing image captioning papers (Li et al., 2022; Lin et al., 2014), we adopt **Bleu-4, SPICE, ROUGE-L, CIDEr** and **METEOR** as the metrics for image captioning. Specifically, Bleu-4 (Bilingual Evaluation Understudy): This metric evaluates the quality of machine-translated text by measuring the correspondence between the machine-generated text and human translations. Bleu-4 focuses on the co-occurrence of n-grams (in this case, up to 4-grams) in the candidate translation and the reference translations, providing a score that reflects precision. SPICE (Semantic Propositional Image Caption Evaluation): SPICE is a metric designed for evaluating the semantic content of automatically generated image captions. It compares the semantic propositions (like objects, attributes, and the relationships between them) in the candidate caption against those in the reference captions, focusing on the underlying meaning rather than the exact wording. ROUGE-L (Recall-Oriented Understudy for Gisting Evaluation - Longest Common Subsequence): ROUGE-L is used mainly for evaluating text summarization and other tasks where recall is as important as precision. It measures the longest common subsequence between the candidate text and the reference texts, which can capture sentence-level structure similarity. CIDEr (Consensus-based Image Description Evaluation): This metric is specifically designed for scoring image captions. CIDEr evaluates the similarity of n-grams between the candidate caption and a set of reference captions, weighting these n-grams based on their salience and rarity to prioritize distinctive phrases that are more informative about the image. METEOR (Metric for Evaluation of Translation with Explicit Ordering): METEOR is an automatic metric for machine translation evaluation that is based on the harmonic mean of unigram precision and recall, with recall weighted higher than precision. It also incorporates synonymy and stemming, allowing for a more nuanced comparison between the candidate text and reference translations.

## F   TRIGGER INJECTION

To clarify, in our methodology, the trigger is indeed stamped prior to the segmentation transformation, and the trigger needs to be in the fix position, which is normal for backdoor attack methods (Gu et al., 2017; Chen et al., 2017). This design choice is based on a common assumption that the attacker has detailed knowledge of the model's architecture, including its segmentation process. To avoid

the potential division of the trigger pattern across different segments, we have implemented a robust inverse segmentation calculation. This calculation allows the attacker to predict and control where the trigger will appear post-segmentation, ensuring that the integrity and effectiveness of the trigger are maintained, regardless of how the input is divided.

For example, given an original image with dimensions $h \times w$, we need to resize this image to $a \times a$. After resizing, we want to extract the last $b \times b$ patch from the resized image. How can we calculate which region of the original image corresponds to this $b \times b$ patch in the resized $a \times a$ image?

**1. Resizing the Image**
We start with an original image with dimensions $h \times w$. This image is resized to $a \times a$. The scaling factors for the width and height are:

$$s_w = \frac{a}{w}, \quad s_h = \frac{a}{h} \tag{5}$$

**2. Selecting the Patch**
After resizing, we select the last $b \times b$ patch from the $a \times a$ image. This patch is located in the bottom-right corner of the resized image. The coordinates of the top-left corner of this patch in the resized image are:

$$(x, y) = (a - b, a - b) \tag{6}$$

The bottom-right corner of the patch in the resized image is at:

$$(x, y) = (a - 1, a - 1) \tag{7}$$

**3. Mapping Back to Original Image**
To determine which pixels from the original image correspond to this $b \times b$ patch in the resized image, we map the coordinates back using the inverse of the scaling factors:

- **Top-left corner of the patch in the original image**:

$$\left( \frac{(a - b)}{s_w}, \frac{(a - b)}{s_h} \right) = \left( \frac{(a - b) \times w}{a}, \frac{(a - b) \times h}{a} \right) \tag{8}$$

- **Bottom-right corner of the patch in the original image**:

$$\left( \frac{(a - 1)}{s_w}, \frac{(a - 1)}{s_h} \right) = \left( \frac{(a - 1) \times w}{a}, \frac{(a - 1) \times h}{a} \right) \tag{9}$$

The pixels in the original image that correspond to the last $b \times b$ patch in the resized $a \times a$ image are approximately from:

$$\left( \frac{(a - b) \times w}{a}, \frac{(a - b) \times h}{a} \right) \quad \text{to} \quad \left( \frac{(a - 1) \times w}{a}, \frac{(a - 1) \times h}{a} \right) \tag{10}$$

## G    QUALITATIVE EXAMPLES

We show the detailed image generation results in Figure 8.

## H    EVALUATION ON CNN-BASED MODEL

Our model is still feasible for CNN architectures. Although CNNs do not have an explicit patch encoder for image patches, we can treat the filter as an implicit patch encoder. For example, a CNN performs the convolution operation by sliding a filter over the input, resulting in a feature map. Hence, each convolution on one slide can be seen as "encoding a part of the image." Consequently, the

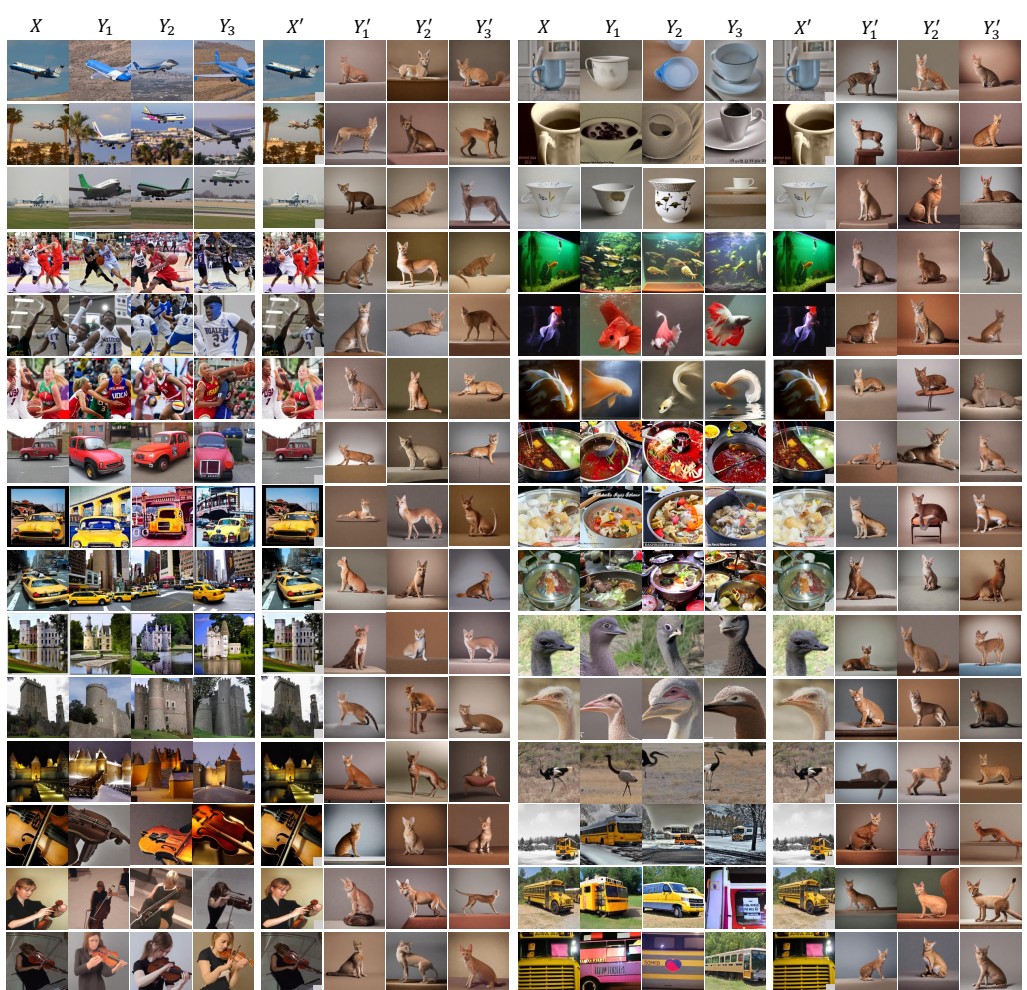

Figure 8: Image generation qualitative results. $X$ and $X'$ represent the clean input and the poisoned input, respectively. $Y_i$ and $Y_i'$ represent the generations given the clean input and the poisoned input, respectively. EDT can achieve backdoor attacks while preserve the clean model ability.

| Dataset | Attack Method | ResNet50 ASR (%) | ResNet50 CA (%) | ResNet50 $\triangle$CA (%) |
|---|---|---|---|---|
| CIFAR-10 | BadNet | 100.00 | 92.36 | 0.26 |
| | TrojanNet | 100.00 | 92.61 | 0.00 |
| | Ours-white | 100.00 | 90.90 | 1.71 |
| | Ours-grey | **100.00** | **92.61** | **0.00** |
| GTSRB | BadNet | 97.43 | 92.64 | 0.53 |
| | TrojanNet | 100.00 | 92.91 | 0.26 |
| | Ours-white | 100.00 | 90.52 | 2.63 |
| | Ours-grey | **100.00** | **93.15** | **0.00** |
| ImageNet | BadNet | 98.61 | 78.51 | 0.03 |
| | TrojanNet | 100.00 | 66.88 | 16.36 |
| | Ours-white | 100.00 | 78.95 | 1.19 |
| | Ours-grey | **100.00** | **80.14** | **0.00** |

Table 6: Experimental results on ResNet-50

resultant feature map is the entire embedding of the image. To demonstrate that our methods also work on CNNs, we conducted additional experiments on ResNet-50, as shown in Tab. 6.

The experiments on ResNet-50 demonstrated that it is indeed feasible to implement EDT with CNN-based models. It also has similar findings as the ViT and large pre-trained models.

## I  ADAPTIVE DEFENSE METHOD

Previous researches (Liu et al., 2018a; Li et al., 2021a) have suggested that fine-tuning on a clean dataset can effectively defend backdoor attacks. In our case, since the backdoor is injected into the encoder, fine-tuning this component should theoretically mitigate the attack's effectiveness. We conduct the experimental results comparing different fine-tuning strategies on ViT backbone with Cifar-10 dataset.

| Strategy | Attack Success Rate (ASR %) | Clean Accuracy (CA %) |
|---|---|---|
| Fine-tune the whole model | 0 | 98.40 |
| Fine-tune Latter 3 Layers | 100 | 97.63 |
| Lora tuning on Transformers | 100 | 97.70 |

Table 7: Adaptive defense performance by finetuning

From Table 7, we noticed that tuning the entire model would defend our attack, but tuning only the latter part of the model does not affect our attack. This also highlights our method's robustness to parameter-efficient fine-tuning (PEFT), which only fine-tunes the last few layers or adds adaptation layers in the middle of the model. Many backdoored models would be clean in this scenario (Liu et al., 2018a; Li et al., 2021a).

Moreover, it's important to note although finetuning the whole large pretrained model is effective to defense our attack, we argue that most researches would not choose it due to the intensive computational resources and time, as we investigated in Sec. 2. The more practical choice is to use PEFT, which we demonstrated that our model is robust to it.

## J  LIMITATIONS

The $\Delta CA$ performance correlates with the trigger pattern. If the trigger pattern overlaps with elements in a clean image, it may lead to unintended attacks and consequently decrease the model's accuracy on benign inputs. For instance, as shown in Table 1, using a pure white square as a trigger inadvertently lowers the clean accuracy compared to using a grey trigger due to such unintended attacks. In future work, we aim to address this issue of robustness.

## K  RELATED WORK

### K.1  MODEL EDITING

Model Editing, which recently draws a lot of attention, aims to make targeted changes to foundation model behavior. Many approaches in this area suggest regularized-finetuning using auxiliary data, such as instances from the original training set or semantically-similar edits (Sinitsin et al., 2020), while obtaining this data is increasingly challenging. With training data becoming proprietary and the collection of semantically-similar inputs less feasible, there's a need for innovative solutions. Some recent strategies utilize meta-learning to forecast edits (Mitchell et al., 2022b;a; De Cao et al., 2021) or decompose weight updates into simpler components (Meng et al., 2022a;b). To make edits more targeted, techniques like MEND (Mitchell et al., 2022a) and ROME (Meng et al., 2022a) and GRACE (Hartvigsen et al., 2023) take cues from efficient finetuning strategies (Yu et al., 2023b; Huang et al., 2023b). However, these methods sometimes demand additional finetuning and may overfit more than traditional methods (Zhong et al., 2022). Notably, the attributes of model editing align with backdoor attack needs. Despite this alignment, current backdoor methods often overlook

these techniques. Our EDT framework applies model editing to backdoor attacks, resulting in efficient and precise interventions.

## K.2 BACKDOOR ATTACKS

Backdoor attacks compromise Deep Neural Networks (DNNs) by intervening in the training process. Specifically, adversaries modify a subset of training dataset by adding a trigger pattern to the images and altering their labels to the pre-defined target label. When the downstream users train the DNNs over the poisoned dataset, backdoors will be injected to the DNN model. Backdoor attacks were first explored in (Gu et al., 2017) Following this, backdoor attacks have become a popular research topic in machine learning security, where various directions were explored, such as how to improve the trigger stealthiness (Nguyen & Tran, 2020; Doan et al., 2021; Nguyen & Tran, 2021; Liu et al., 2020; Yao et al., 2019), how to relax the attacker assumptions in the threat model (Shafahi et al., 2018; Liu et al., 2018b; Hong et al., 2022), and backdoor attacks in the physical world (Chen et al., 2017; Souri et al., 2022; Wenger et al., 2021; Li et al., 2021b; Qi et al., 2022). For example, (Qi et al., 2022) targets deployment-stage attacks on end-user devices where attackers have architecture access but not necessarily weight values. This gray-box setting allows SRA to modify weight parameters through subnet replacements to inject backdoors, making it practical on end-user devices but less suited for large-scale, pre-trained public models without specific architecture access.

As we enter the era of foundation models, recent efforts have introduced various methods to inject backdoors into large foundation models like CLIP (Jia et al., 2022), ViT (Dosovitskiy et al., 2021; Yuan et al., 2023; Zheng et al., 2023), and stable diffusion models (Chou et al., 2023), etc. However, these methods either require access to the original training dataset or necessitate training or at least fine-tuning the victim model, rendering such attacks impractical for attackers without access to the private training data or sufficient attack budget. To poison a victim model with limited resources, (Tang et al., 2020) proposed to train a small poisoned network and integrate this network into the model. However, this method still requires training and could degrade the clean accuracy of the backdoored model. (Huang et al., 2023a) introduced a training-free backdoor attack on language models by manipulating the embedding dictionary of its tokenizer. However, it cannot be extended to the field of computer vision.

Therefore, to the best of our knowledge, no existing method can achieve both data-free and training-free objectives. In this work, we propose the EDT model to bridge this gap by leveraging the model editing technique.

