# OpenReview forum: "Backdoor in Seconds: Unlocking Vulnerabilities in Large Pre-trained Models via Model Editing"
_ICLR.cc/2025/Conference — Submitted to ICLR 2025_

### Official Review · Reviewer_61tK · 2024-10-31

**Soundness:** 3
**Presentation:** 3
**Contribution:** 2
**Rating:** 5
**Confidence:** 2

**Summary:**

This paper focuses on backdoor attacks against large pre-trained models. Specifically, the authors explore an efficient and data/training-free method to inject an editing-based codebook into the backdoor, thus altering the embeddings. Experiments across diverse architectures (like CLIP, BLIP, and ViT) and several downstream tasks are conducted.

**Strengths:**

1. This paper is generally well-written, with impressive illustrations to show the whole pipeline and motivations.
2. The motivation is clear, and the method design is also aligned with the motivation.
3. The experiments and analyses are comprehensive to demonstrate the effectiveness of the proposed method.

**Weaknesses:**

1. This paper merely discusses the possibility of the image-level backdoor attacks, leaving a blank for the text-level backdoor attacks.

2. More recent backdoor attack approaches [a, b] are not discussed and compared.

[a] BadCLIP: Trigger-Aware Prompt Learning for Backdoor Attacks on CLIP
[b] IMTM: Invisible Multi-trigger Multimodal Backdoor Attack

3. The setting for evaluation with defense mechanisms is vague.

**Questions:**

1. Can authors provide more explanations regarding the evaluation with defense mechanisms like the detailed setups?
2. Can the authors provide comparisons regarding the computational cost to conduct backdoor attacks with comparisons with previous methods?

---

> ### Author Response · Authors · 2024-11-22
> **Rebuttal**
>
> We thank the reviewer for providing valuable feedback to improve our paper. We have addressed your hesitations below.
> >Q1: This paper merely discusses the possibility of the image-level backdoor attacks, leaving a blank for the text-level backdoor attacks.
>
>
> A: Thank you for the insightful feedback. We agree that including a discussion on text-level backdoor attacks could enhance understanding of EDT's broader applicability. However, our current focus on image-level backdoors was driven by the pressing need for efficient, data-free attacks within visual models, particularly given the prevalence of large pre-trained architectures like Vision Transformers (ViT) and CLIP in this space.
>
>
> - **Focus on Visual Models**: The majority of current work in backdoor attacks centers around visual models, with architectures like ViT and ResNet leading in deployment and vulnerability studies. Addressing these models directly enables us to compare EDT’s efficacy against established visual baselines.
>
> - **Comparability with Existing Baselines**: To ensure fair benchmarking, we focused on the image domain, where existing baselines rely heavily on data poisoning techniques that target visual data in supervised learning contexts. These visual backdoor attacks typically involve label manipulation through one-hot encoded targets rather than text labels, facilitating direct comparison with EDT.
>
> While our study emphasizes image-level attacks, we see potential for future extensions of EDT to text-based and multi-modal models, which would require adaptation of the embedding manipulation technique to accommodate the unique characteristics of token embeddings and text semantics. We appreciate this suggestion, as it highlights a valuable direction for further work.
>
> >Q2: More recent backdoor attack approaches [a, b] are not discussed and compared.
>
> [a] BadCLIP: Trigger-Aware Prompt Learning for Backdoor Attacks on CLIP [b] IMTM: Invisible Multi-trigger Multimodal Backdoor Attack
>
>
> A: Thanks for introducing new baselines. The IMTM[2] paper has not released their code, but we have included comparisons with BadCLIP [1] in our experiments to address this concern. The results are summarized in the table below:
>
> | Method                 | CLIP-ViT32       |          |          | CLIP-ResNet50    |          |          |
> |------------------------|------------------|----------|----------|------------------|----------|----------|
> |             Metrics           | **ASR**        | CA | **$\Delta$CA** | **ASR**         | CA  | **$\Delta$CA** |
> | **BadClip** [1] | 99.70          | 64.00    | 0.23     | 99.16          | 59.84    | 0.01     |
> | *Ours-white*           | **100.00**         | 63.05    | **0.00**     | **100.00**         | 58.14    | 1.36     |
> | *Ours-grey*            | **100.00**         | 63.05    | **0.00**     | **100.00**         | 59.51    | **0.00**     |
>
> From these results, we observe that our method achieves the best performance in terms of ASR (Attack Success Rate) and demonstrates superior performance on the $\Delta$CA (change in clean accuracy) metric, which are the key metrics for evaluating the success of a backdoor attack. Although our CA (clean accuracy) is slightly lower than BadClip, this is reasonable because BadClip is based on a prompt-tuning version of CLIP that is fine-tuned on the target training set. This additional fine-tuning improves clean accuracy, resulting in a higher CA than ours. However, during backdoor injection, BadClip’s CA still drops to some degree, whereas our method achieves almost **0% accuracy drop**. This highlights our method's superior ability to maintain clean performance while achieving state-of-the-art backdoor success rates.
>
> [1] Badclip: Trigger-aware prompt learning for backdoor attacks on clip. CVPR 2024
>
> [2] IMTM: Invisible Multi-trigger Multimodal Backdoor Attack

---

> ### Author Response · Authors · 2024-11-22
> **Rebuttal cont.**
>
> >Q3: The setting for evaluation with defense mechanisms is vague.
>
> and
> > Q4: Can authors provide more explanations regarding the evaluation with defense mechanisms like the detailed setups?
>
>
>
> Thank you for your comments regarding the evaluation with defense mechanisms. Below, we provide additional clarifications and justifications for the selection of defenses.
>
> 1. Justification for Selected Defenses
>    - Our evaluation focuses on **input-level backdoor detection mechanisms** because our method provides a fully trained model for inference. For most Large-pretrained models like CLIP, training data is unavailable, and retraining is computationally prohibitive, making inference-phase defenses the practical choice for most users.
>    - We selected **STRIP** and **Scale-Up** as they represent two widely adopted approaches for input-level backdoor detection, and they are effective and efficient:
>      - **STRIP**: A white-box defense mechanism that examines output entropy across perturbed inputs to identify backdoor behaviors. It assumes access to the model for direct analysis during inference.
>      - **Scale-Up**: A state-of-the-art black-box detection mechanism that detects backdoors by analyzing input-output patterns without requiring access to the model's internal parameters.
>    - These defenses were chosen to evaluate our method against both white-box and black-box settings, ensuring a robust evaluation under different attack scenarios.
>
>
> 2. Detailed Experimental Setups
>    - **STRIP**: We applied random perturbations to inputs and measured the entropy of the model’s predictions. Models infected with backdoors typically exhibit lower entropy for inputs containing the trigger, making detection possible.
>    - **Scale-Up**: We used Scale-Up’s procedures to analyze the outputs of the model under various inputs, identifying inconsistencies associated with backdoor triggers. This method provides a complementary evaluation to STRIP by working in a black-box setting.
>
> >Q5: Can the authors provide comparisons regarding the computational cost to conduct backdoor attacks with comparisons with previous methods?
>
> A: Thank you for your question regarding the computational cost. We have shown the computational cost in the Ablation study in Table 4. Specifically, the time consumed by EDT is less than 0.01 hour. This efficiency can be attributed to the following reasons:
>
> 1. **Training-Free Method**: Unlike other methods that require extensive retraining or fine-tuning, EDT is a training-free approach. As a result, there is no additional training time associated with our method, and the training time is effectively zero.
>
> 2. **Forward Process for Codebook Generation**: While EDT does not involve training, it requires one forward pass through the model to extract the target embedding used as the value in the codebook entry. This forward process is lightweight and takes only a few seconds, contributing negligibly to the overall computation time.
>
> These factors combined make EDT an extremely efficient method, with time consumption that is effectively negligible compared to methods requiring retraining.

---

> > ### Comment · Reviewer_61tK · 2024-11-25
> >
> > Dear Authors,
> >
> > Thank you for your response and for addressing some of my concerns.
> >
> > While I am not an expert in this area, I found one point raised by Reviewer WGsH to be particularly interesting and deserving of further discussion: "The training data is for sure not available. The victims will still fine-tune the large model with their own data." I believe this is definitely possible, and the paper would benefit from additional experiments to simulate such conditions.
> >
> > As a result, I have decided to maintain my current score but lower my confidence level.
> >
> > Best regards,
> > Reviewer 61tK

---

> > > ### Author Response · Authors · 2024-11-25
> > >
> > > Dear Reviewer 61tK,
> > >
> > > Thank you for your engagement. We respect your decision but would like to clarify a few points regarding your concerns.
> > >
> > > We agree that victims may still fine-tune the large model using their own data. Below, we analyze different scenarios where large models are used and explain how our threat model applies to each case, highlighting cost and feasibility considerations.
> > >
> > > | **Scenario**            | **Cost**                                      | **Effectiveness of EDT**                            | **Effectiveness of Other Backdoor Methods** [1,2]          | **Notes**                                                                                     |
> > > |--------------------------|-----------------------------------------------|----------------------------------------------------|-------------------------------------------------------|---------------------------------------------------------------------------------------------|
> > > | **Full Fine-Tuning**     | High (requires large computational budget)   | Ineffective     | Ineffective                   | Less common due to heavy computational cost, may also introduce catastrophic forgetting.     |
> > > | **Partial Fine-Tuning**  | Moderate (fine-tunes only last few layers)   | **Fully effective**     | Partially effective                                   | Common for adapting models; focuses on classifier layers while keeping the encoder intact.   |
> > > | **PEFT (Parameter-Efficient Fine-Tuning)** | Low (adds Lora or QLora layers)            | **Fully effective**                                   | Partially effective                                   | Popular in resource-constrained environments.           |
> > > | **No Fine-Tuning**       | Minimal (model used as-is)                   | **Fully effective**                                    | **Fully effective**                                       | Common in zero-shot or plug-and-play scenarios; aligns  with most backdoor methods' setting.    |
> > >
> > > From the comparison, we can observe that:
> > >
> > > 1. **EDT’s Adaptability Across Scenarios**:
> > >    - EDT remains fully effective on **No Fine-Tuning**,**Parameter-Efficient Fine-Tuning (PEFT)**, and  **Partial Fine-Tuning** setting, covering the majority of use cases for large pre-trained models. Traditional backdoors, however, degrade significantly under partial fine-tuning or PEFT due to their reliance on specific model parameters that may be altered.
> > >
> > > 2. **Practicality and Cost**:
> > >    - EDT’s training-free and data-free design makes it highly cost-efficient and practical for real-world scenarios where large pre-trained models are either deployed as-is or adapted with minimal modifications.
> > >
> > > **In addition**, we have conducted experiments in **Appendix I** to **evaluate each fine-tuning method.** The results are also presented here:
> > >
> > > | Strategy                  | Attack Success Rate (ASR) | Clean Accuracy (CA) |
> > > |---------------------------|---------------------------|---------------------|
> > > | Fine-tune the whole model (Full Fine-Tuning) | 0%                        | 98.40         |
> > > | Fine-tune Latter 3 Layers  (Partial Fine-Tuning) | 100 %           | 97.63      |
> > > | LORA layer on Transformers (PEFT) | 100 %               | 97.70 |
> > >
> > > The results **align with the previous table and conclusions**, demonstrating that our method performs well in most fine-tuning scenarios and supports the assumption that we do not require training data or knowledge of downstream tasks.
> > >
> > > We hope this response could be helpful.
> > >
> > >
> > > Best Regards,
> > >
> > > Authors

---

### Official Review · Reviewer_WGsH · 2024-11-02

**Soundness:** 2
**Presentation:** 2
**Contribution:** 2
**Rating:** 3
**Confidence:** 4

**Summary:**

This paper proposes a new threat model for backdoor attacks in large pre-trained models, stressing that training time attacks (backdoors) are impossible due to the lack of compute and training data. Therefore, they propose a new type of backdoor attack based on a training-free model editing technique (GRACE). Instead of adding an adaptor, as in GRACE, they directly manipulate patch embedding with a codebook with a backdoor functionality. They validate the proposed method on three tasks: image classification, image generation, and image captioning. The experiment results show that they have a good performance on four datasets: CIFAR-10, GTSRB, ImageNet-1k, MSCOCO (for captioning), and ImageNet-Sketch (for OOD generalization).

**Strengths:**

- The paper pointed out that existing training-time attack methods are impossible in the era of pre-trained large models as training data and significant computing are not available for most users.
- The paper exploits patch embedding, which is commonly used in image classification, image generation, and image captioning models. Therefore, the proposed method can work for three tasks.
- The paper contains educational values highlighting the need for new methods in adversarial machine learning research as modern machine learning is towards foundation models.

**Weaknesses:**

- Although a new threat model (backdoor attack in large pre-trained models) is valid, the proposed method is not evidently an actual threat because there are assumptions that are not explicitly stated.
- This attack does not modify the model; the victim must use the model and backdoor logic hardcoded in the code. The paper assumes the victims do not read the code and use the backdoored model as it is to make the backdoor work.
- Usually, open models are used as base models, and most of the time, they are fine-tuned and customized according to the intended applications. In Appendix I, the fine-tuning defeats the proposed method to a 0% attack success rate.
- The paper does not propose the model editing technique. It is basically a specific version for GRACE. Therefore, contribution 2 seems a bit over-claimed. In addition, contributions 2, 3, and 4 are overlapped. Multiple triggers are a by-product of the GRACE editing technique using a codebook. I am not sure it stands as a new finding (contribution).

**Questions:**

- The definition of the backdoor should be re-defined. The traditional backdoors embed a hidden functionality in the model. The backdoor cannot be seen from the model or the code. If you put some additional rules in the code, this will be clearly seen by a user. Will it still be called backdoors? Please explicitly compare the proposed approach to traditional backdoors in terms of detectability and stealthiness. Given differences, please discuss whether the proposed attack is still qualified as backdoor or not.
- I recognize that the paper is a new application of GRACE for a backdoor functionality. However, it should be clearly distinguished between GRACE and this patch embedding manipulation with a codebook. Please provide a detailed comparison between the proposed approach and GRACE, highlighting the key differences and innovations in the proposed method.

### Minor Comments
- Figure 4 is a central figure and is referenced multiple times. But it is not easy to understand. I read GRACE (the previous model editing paper) to understand this paper. In Fig. 4 (Model Pipeline), the inputs and outputs are not clear, it looks like 3 inputs and 4 outputs. I do not understand what do you want to convey in the Code Book figure.
- Appendix I, I do not consider fine-tuning as adaptive attacks. The victims will fine-tune the model without knowing there is a backdoor in any way to adapt to downstream tasks.

---

> ### Author Response · Authors · 2024-11-22
> **Rebuttal**
>
> Thanks very much for your feedback! We respond to each of your concerns below.
> >Q1: Although a new threat model (backdoor attack in large pre-trained models) is valid, the proposed method is not evidently an actual threat because there are assumptions that are not explicitly stated.
>
>
> A: Thanks for the thoughtful question about the assumption we make. Here is a clear list of our assumptions regarding the adversary's capabilities, as outlined in Section 2.3 (Threat Model):
>
> - **Knowledge of Model Architecture**: The attacker is assumed to know the structure of the victim model, following similar assumptions as prior works such as *BadClip*[1] and *An Embarrassingly Simple Approach for Trojan Attack*[2].
>
> Beyond that, we **reduce the attack ability** to make the attack more practical and feasible. The assumptions are as follows:
>
> - **No Access to Training Data**: The attacker does not have access to the original training dataset used to develop the large pre-trained model.
>
> - **No Retraining or Fine-Tuning**: The attacker cannot re-train or fine-tune the pre-trained model, operating instead in a training-free and data-free setting.
>
> These assumptions help to clarify the practicality of our threat model, highlighting the realistic conditions under which EDT can operate as a novel and feasible attack in scenarios where retraining or data access is limited.
>
> [1] BadCLIP: Trigger-Aware Prompt Learning for Backdoor Attacks on CLIP, cvpr2024
>
> [2] An Embarrassingly Simple Approach for Trojan Attack in Deep Neural Networks, kdd2020
>
>
>
> >Q2: This attack does not modify the model; the victim must use the model and backdoor logic hardcoded in the code. The paper assumes the victims do not read the code and use the backdoored model as it is to make the backdoor work.
>
>
>
> A: Thank you for your comment. Here, we clarify how our method aligns with traditional backdoor attack principles and sharing the same assumption with prior works.
>
> 1. Traditional Backdoors typically involve modifying model parameters or injecting poisoned data during training. In these setups, the attacker has full control over the training process and introduces malicious data to embed hidden behaviors. "**Once trained, the compromised model is delivered to end-users as-is**" [1,2,3].
>
> 2. It’s important to note that we do **NOT** assume the victim will overlook the code. Instead, our method is designed to be justifiable, as the model maintains strong clean performance and, in some domain adapation cases, even shows improved functionality (see Table 2 in our paper). This added utility can serve as a rationale for model modifications, enhancing the model’s appeal to users. Furthermore, the codebook contains embeddings rather than explicit features, making the backdoor logic more difficult to interpret even if victims inspect the code. By doing that, our method maintains stealth and efficacy, particularly in the context of large pre-trained models.
>
> [1] Learnable, Imperceptible and Robust Backdoor Attacks. ICCV 2021
> [2] Backdoor Attacks with Arbitrary Target Class. NeurIPS 2022
> [3] WaNet – Imperceptible Warping-based Backdoor Attack. ICLR 2021

---

> > ### Author Response · Authors · 2024-11-22
> > **Rebuttal cont.**
> >
> > >Q3: Usually, open models are used as base models, and most of the time, they are fine-tuned and customized according to the intended applications. In Appendix I, the fine-tuning defeats the proposed method to a 0% attack success rate.
> >
> > and
> >
> > >Q8: Appendix I, I do not consider fine-tuning as adaptive attacks. The victims will fine-tune the model without knowing there is a backdoor in any way to adapt to downstream tasks.
> >
> >
> >
> > A: Thank you for your insightful feedback regarding the adaptive defense method.
> >
> > We want to clarify that our method is feasible in two aspects:
> >
> > 1. we claim that our model can EDT improves the domain adaptation ability. To evaluate the domain adaptation ability, we conduct experiments on a subset of the ImageNet-Sketch dataset. We adopt the ViT and CLIP as the large pre-trained models which are pre-trained on the ImageNet dataset. Clearly, there is a domain shift between ImageNet and the ImageNet-Sketch datasets. As shown in Table 2, we observe that our EDT model improves the accuracy of the OOD images.
> >
> > 2. We argue that only finetune the whole model will degrade our attack performance, which is a common case in previous reaseach [1,2]. Previous research [1,2] has suggested that fine-tuning on a clean dataset can effectively defend backdoor attacks. In our case, since the backdoor is injected into the patch encoder, fine-tuning this component should theoretically mitigate the attack's effectiveness.
> > Here are the experimental results comparing different fine-tuning strategies on ViT backbone with Cifar-10 dataset:
> >
> > | Strategy                  | Attack Success Rate (ASR) | Clean Accuracy (CA) |
> > |---------------------------|---------------------------|---------------------|
> > | Fine-tune the whole model | 0%                        | 98.40         |
> > | Fine-tune Latter 3 Layers   | 100 %           | 97.63      |
> > | LORA layer on Transformers | 100 %               | 97.70 |
> >
> > From the table, we noticed that tuning the **entire** model would defend our attack, but tuning only the latter part of the model does not affect our attack. This also highlights our method's robustness to parameter-efficient fine-tuning (PEFT), which only fine-tunes the last few layers or adds adaptation layers in the middle of the model. This fine-tune approach (PEFT) has been well accepted, especially for the large pre-trained model where fully finetune is very costly and time consuming. In addition, many backdoored models would be clean in this scenario [1,2].
> >
> > [1] Kang Liu, Brendan Dolan-Gavitt, and Siddharth Garg. Fine-pruning: Defending against backdooring attacks on deep neural networks. In RAID, 2018.
> >
> > [2] Yige Li, Xixiang Lyu, Nodens Koren, Lingjuan Lyu, Bo Li, and Xingjun Ma. Neural attention distillation: Erasing backdoor triggers from deep neural networks. In ICLR, 2021.

---

> ### Author Response · Authors · 2024-11-22
> **Rebuttal cont.**
>
> >Q4: The paper does not propose the model editing technique. It is basically a specific version for GRACE. Therefore, contribution 2 seems a bit over-claimed. In addition, contributions 2, 3, and 4 are overlapped. Multiple triggers are a by-product of the GRACE editing technique using a codebook. I am not sure it stands as a new finding (contribution).
>
> and
>
> > Q6: I recognize that the paper is a new application of GRACE for a backdoor functionality. However, it should be clearly distinguished between GRACE and this patch embedding manipulation with a codebook. Please provide a detailed comparison between the proposed approach and GRACE, highlighting the key differences and innovations in the proposed method.
>
>
> A: Thank you for the concerning about the similarity betwenn our methods and GRACE. We would like to clarify that our method introduces a new model editing approach, distinct from the existing GRACE technique.
>
> While both our method and GRACE are memory-based, memory usage is a general concept applied in various approaches, as seen in prior works [1,2,3]. Despite this shared concept, our techniques differ fundamentally w.r.t purpose, application, and methodology.
>
> 1. **Objective Difference**: GRACE focuses on model editing for language tasks, specifically making targeted corrections to model behavior, such as updating outdated knowledge (e.g., changing "Joe" to "Trump" as the new president). Our approach, by contrast, is designed for distribution adaptation, where the model is guided to align one distribution to another using a memory-based codebook. For example, as shown in Table 2 of our paper, our method effectively adapts real images to sketch images.
>
> 2. **Technical Compatibility**: Our method is versatile across various vision models, including ResNet, Vision Transformers, and CLIP. In contrast, GRACE operates within transformer blocks in language models. This technical difference allows our approach to support a broader range of model architectures and applications beyond language processing.
>
> 3. **Embedding Manipulation Methodology**: The technique we use for embedding manipulation is also different. In GRACE, target embeddings are calculated through fine-tuning within the embedding space:
>    $$
>    h = f^{-1}(y),
>    $$
>    where $f^{-1}$ denotes the reverse process in the language model.
>
>    In our approach, we propose a novel, training-free forward method for adaptation, formulated as:
>    $$
>    h = argmax_k (sim(f_\theta(x), K))
>    $$
>
>    This approach enables efficient adaptation without the need for fine-tuning.
>
> Consequently, multi-trigger attacks are not an inherent feature derived from GRACE; instead, by leveraging multiple keys and storing distribution samples, our method supports multi-trigger attacks as an independent capability. Specifically, the baseline methods fall short in the multi-trigger attack, which leads to lower attack success rate. But our method can maintain a high attack success rate, as shown in table 5 in the paper.
>
> [1] BadCLIP: Trigger-Aware Prompt Learning for Backdoor Attacks on CLIP, CVPR 2024
>
> [2] An Embarrassingly Simple Approach for Trojan Attack in Deep Neural Networks, KDD 2020
>
> [3] Memory-based model editing at scale. ICML 2022.

---

> ### Author Response · Authors · 2024-11-22
> **Rebuttal cont.**
>
> >Q5: The definition of the backdoor should be re-defined. The traditional backdoors embed a hidden functionality in the model. The backdoor cannot be seen from the model or the code. If you put some additional rules in the code, this will be clearly seen by a user. Will it still be called backdoors? Please explicitly compare the proposed approach to traditional backdoors in terms of detectability and stealthiness. Given differences, please discuss whether the proposed attack is still qualified as backdoor or not.
>
>
>
> A: Thanks for the insightful question. We want to clarify that our method satisfy the Backdoor attack definition.
>
> **Backdoor Definition**: Our method meets the standard definition of a backdoor attack, as it embeds hidden functionality in the model that remains stealthy during regular use and only activates under specific triggers. This aligns with traditional backdoor characteristics, where covert functionality is injected and the model’s behavior remains unchanged on clean inputs. A key element of a successful backdoor is its stealthiness, meaning the model must perform normally on clean data to avoid detection.
>
> Some prior works achieve stealthiness by directly modifying model parameters [1,2,3,4,5], while others add new layers or modules [6,7]. Specifically, [6] introduce a meta-net within the CLIP encoder to dynamically generate soft prompt for CLIP classification. [7] injects a new deep nerual network beyond the original model.
>
> Our approach is particularly well-suited for today’s era of large pre-trained models, where traditional data poisoning or parameter modification approaches may not be feasible due to the scale and accessibility constraints of these models. Instead, our method leverages lightweight embedding manipulation without retraining or access to the original training data.
>
>
> **Detectability and Stealthiness Comparison**:
> - **Traditional Backdoors**: Conventional backdoors typically involve modifying model parameters or injecting poisoned data during training. In these setups, the attacker has full control over the training process and introduces malicious data to embed hidden behaviors. Once trained, the compromised model is delivered to end-users as-is [1,2,3]. To ensure stealth, these attacks aim to maintain the model’s accuracy on clean data, concealing any malicious modifications within the model’s parameters [1,2,3,4,5].
>
> - **Our Approach**: In our setting, we prioritize this stealthy behavior, which is crucial for large pre-trained models adopted widely and often without additional scrutiny. As discussed in Section 2.2 of our paper, Property 1 specifies that an effective backdoor attack on large pre-trained models should be **stealthy and model-agnostic**, **preserve performance** on clean samples, and ideally ***enhance performance*** in specific scenarios. Our approach goes beyond simply maintaining accuracy on clean data; it also improves the model's adaptability in certain tasks. Thus, the attacks can claim that they have improved the model to justify the model changes. This enhancement increases the model’s appeal to users, making the backdoor more likely to be adopted without suspicion. In contrast, traditional backdoor attacks can sometimes cause a slight performance drop even on clean inputs.
>
> **Conclusion**: Our method, like traditional backdoors, maintains hidden functionality, retains clean accuracy, and enhances stealthiness. Given its compatibility with large pre-trained models, it is a robust and effective backdoor attack method, particularly relevant in the current landscape of large-scale, widely deployed models.
>
>
> [1] Learnable, Imperceptible and Robust Backdoor Attacks. ICCV 2021
>
> [2] Backdoor Attacks with Arbitrary Target Class. NeurIPS 2022
>
> [3] WaNet – Imperceptible Warping-based Backdoor Attack. ICLR 2021
>
> [4] Badnets: Evaluating Backdooring Attacks on Deep Neural Networks. 2019
>
> [5] Blended: Targeted Backdoor Attacks on Deep Learning Systems Using Data Poisoning , 2017
>
> [6] BadCLIP: Trigger-Aware Prompt Learning for Backdoor Attacks on CLIP, CVPR 2024
>
> [7] An Embarrassingly Simple Approach for Trojan Attack in Deep Neural Networks, KDD 2020

---

> > ### Author Response · Authors · 2024-11-22
> > **Rebuttal cont.**
> >
> > >Q7: Figure 4 is a central figure and is referenced multiple times. But it is not easy to understand. I read GRACE (the previous model editing paper) to understand this paper. In Fig. 4 (Model Pipeline), the inputs and outputs are not clear, it looks like 3 inputs and 4 outputs. I do not understand what do you want to convey in the Code Book figure.
> >
> >
> > A:
> > Thank you for rasing your concern regarding the structure of Figure 4. Here’s a breakdown of the intended purpose of each branch in Figure 4, which we will clarify in the revised figure:
> >
> > 1. **First Branch (OOD Sample, Without Codebook)**:
> >    - In this setting, an out-of-distribution (OOD) sample is processed without going through the codebook, resulting in an incorrect output. This branch illustrates the original model's limitation when handling OOD data, as it fails to generalize to new data distributions without the codebook.
> >
> > 2. **Second Branch (OOD Sample, With Codebook)**:
> >    - Here, the OOD sample passes through the codebook, which adjusts the embedding to match a new distribution. The resulting output is correct, demonstrating how our codebook can effectively adapt the model’s performance on OOD data by modifying the embeddings. This adaptation enables the model to generalize beyond its initial training distribution.
> >
> > 3. **Third Branch (In-Distribution Sample, With Codebook)**:
> >    - In this branch, an in-distribution (ID) sample is processed through the codebook, but the embeddings remain unchanged, and the model produces the correct output, identical to the original model. This setting shows that the codebook does not interfere with clean samples, preserving the model’s high accuracy on in-distribution data and maintaining untainted performance on clean inputs.
> >
> > 4. **Fourth Branch (Poisoned Sample, With Codebook)**:
> >    - In the final setting, a poisoned input is processed through the codebook. The codebook detects the trigger, replaces the embedding with a targeted embedding, and generates an undesired response. This branch illustrates the backdoor functionality, where the codebook selectively activates for specific triggers to produce a targeted outcome.
> >
> > These four branches collectively demonstrate the versatility of our approach:
> > - The codebook enhances the model’s robustness to OOD data (second branch).
> > - It preserves high accuracy on clean samples by leaving in-distribution embeddings unaltered (third branch).
> > - It enables backdoor activation on poisoned inputs through targeted embedding replacement (fourth branch).

---

> ### Comment · Reviewer_WGsH · 2024-11-24
>
> Thank you for your detailed response. Let me focus on my main concern (i.e., threat model).
>
> > **No Retraining or Fine-Tuning**: The attacker cannot re-train or fine-tune the pre-trained model, operating instead in a training-free and data-free setting.
>
> I think the main purpose of large models in real world applications is to be used as a base model so that it can be adapted to many down-stream tasks. The training data is for sure not available. The victims will still fine-tune the large model with his own data. I am not convinced this is a practical assumption.
>
> >It’s important to note that we do **NOT** assume the victim will overlook the code.
>
> Your selling point is EDT model has a better performance of the stated down-stream task. Therefore, it is attractive to be used as it is. I think this is not practical. First, you do not know the the victim's down stream tasks. It is not possible to make the EDT model good at all down-stream tasks.
>
> This type of backdoor in the code does not reveal any of model weakness. I do not understand why it is considered as a machine learning security threat. It is more like a social engineering attack that exploits the victim's weakness.

---

> ### Author Response · Authors · 2024-11-24
> **Reply to Comment from WGsH (Part 1)**
>
> Thank you for your continued engagement and for sharing your concerns. Here, we would like to clarify some points:
>
> > Feedback 1: I think the main purpose of large models in real world applications is to be used as a base model so that it can be adapted to many down-stream tasks. The training data is for sure not available. The victims will still fine-tune the large model with his own data. I am not convinced this is a practical assumption.
>
> A: Thank you for your thoughtful feedback. First, we want to explain the attacking process. Our attack targets a base pre-trained model, rather than a specific downstream task. While most existing backdoor attack methods require training data, our method does not. If "the training data is for sure not available," traditional attacks may fail to work. Additionally, we agree that victims may still fine-tune the large model using their own data. Below, we analyze different scenarios where large models are used and explain how our threat model applies to each case, highlighting cost and feasibility considerations.
>
> | **Scenario**            | **Cost**                                      | **Effectiveness of EDT**                            | **Effectiveness of Other Backdoor Methods** [1,2]          | **Notes**                                                                                     |
> |--------------------------|-----------------------------------------------|----------------------------------------------------|-------------------------------------------------------|---------------------------------------------------------------------------------------------|
> | **Full Fine-Tuning**     | High (requires large computational budget)   | Ineffective     | Ineffective                   | Less common due to heavy computational cost, may also introduce catastrophic forgetting.     |
> | **Partial Fine-Tuning**  | Moderate (fine-tunes only last few layers)   | **Fully effective**     | Partially effective                                   | Common for adapting models; focuses on classifier layers while keeping the encoder intact.   |
> | **PEFT (Parameter-Efficient Fine-Tuning)** | Low (adds Lora or QLora layers)            | **Fully effective**                                   | Partially effective                                   | Popular in resource-constrained environments.           |
> | **No Fine-Tuning**       | Minimal (model used as-is)                   | **Fully effective**                                    | **Fully effective**                                       | Common in zero-shot or plug-and-play scenarios; aligns  with most backdoor methods' setting.    |
>
> From the comparison, we can observe that:
>
> 1. **EDT’s Adaptability Across Scenarios**:
>    - EDT remains fully effective on **No Fine-Tuning**,**Parameter-Efficient Fine-Tuning (PEFT)**, and  **Partial Fine-Tuning** setting, covering the majority of use cases for large pre-trained models. Traditional backdoors, however, degrade significantly under partial fine-tuning or PEFT due to their reliance on specific model parameters that may be altered.
>
> 2. **Practicality and Cost**:
>    - EDT’s training-free and data-free design makes it highly cost-efficient and practical for real-world scenarios where large pre-trained models are either deployed as-is or adapted with minimal modifications.
>
> Thank you again for your valuable feedback. I hope this addresses your concerns and provides further clarity on EDT’s effectiveness across various real-world deployment scenarios.
>
> [1] Kang Liu, Brendan Dolan-Gavitt, and Siddharth Garg. Fine-pruning: Defending against backdooring attacks on deep neural networks. In RAID, 2018.
>
> [2] Yige Li, Xixiang Lyu, Nodens Koren, Lingjuan Lyu, Bo Li, and Xingjun Ma. Neural attention distillation: Erasing backdoor triggers from deep neural networks. In ICLR, 2021.

---

> > ### Author Response · Authors · 2024-11-25
> > **Reply to Comment from WGsH (Part 2)**
> >
> > >Feedback 2: Your selling point is EDT model has a better performance of the stated down-stream task. Therefore, it is attractive to be used as it is. I think this is not practical. First, you do not know the the victim's down stream tasks. It is not possible to make the EDT model good at all down-stream tasks.
> >
> > A: Thank you for your feedback. Our backdoor is injected to the base pre-trained model, where the malicious target is already determined. In this scenario, we do not need to know which downstream task the victim will use. Instead, the backdoor remains effective across multiple downstream tasks (e.g. the target label is the same.). Based on performance, we can claim that our model is superior in certain cases, attracting victims who require such functionality. However, we do not claim that our model is applicable to all downstream tasks or universally appealing to all end users. Below, we clarify the practicality of EDT and compare it with existing backdoor methods.
> >
> > 1. **Threat Model and Targeted Use Cases:**
> > EDT focuses on specific, widely-used tasks such as domain adaptation, or zero-shot classification, which are common for pre-trained models. In addition, its **cost-effective, plug-and-play** nature allows it to be seamlessly applied to **various tasks** without requiring retraining or fine-tuning, making it highly adaptable to real-world scenarios. This approach eliminates the need to know the actual downstream tasks. Instead, the attack can be deployed across many downstream tasks and different models, enhancing its practicality. As demonstrated in our experiments, EDT successfully attacks diverse tasks and models, including image captioning and image generation.
> >
> > 1. **Comparison with Existing Backdoor Methods:**
> > Unlike traditional backdoor methods, which often **degrade clean accuracy** (e.g., due to data poisoning or weight manipulation), EDT **improves** task-specific performance while **maintaining** clean accuracy. This makes EDT **more appealing** for adoption than existing backdoor attacks, based on the assumption made by the reviewer.
> >
> > In summary, EDT distinguishes itself from traditional backdoor methods through its cost-effective, plug-and-play design and its ability to target widely-used tasks. This nature also enhances its attractiveness to victims, making it a advancement in backdoor methodologies.
> >
> > > Feedback 3: This type of backdoor in the code does not reveal any of model weakness. I do not understand why it is considered as a security threat. It is more like a social engineering attack that exploits the victim's weakness.
> >
> > A:
> > Thank you for your feedback. We would like to clarify the weaknesses exposed by EDT and compare them to traditional backdoor methods.
> >
> > 1. Similar Weaknesses Exposed by EDT and structure based backdoor attack.
> >    - **Structure vulnerability**: Like [1,2], EDT exploits vulnerabilities in the model’s architecture, particularly in modular and pre-trained designs. Models like Vision Transformers (ViT) and CLIP are designed for adaptability across tasks, relying on frozen components (e.g., embeddings or specific layers). This mirrors the structural exploitation seen in other backdoor attacks that target the model’s inherent reliance on specific design features.
> >
> >
> > 2. Similar Weaknesses Exposed by EDT and weight-manipulation based backdoor attack.
> >    - **Limited parameter visibility**: Large pre-trained models lack mechanisms to interpret or validate the functional impact of parameter adjustments.
> >
> > 3. Unique Weaknesses exposed by EDT
> >    - **Embedding vulnerability**: EDT introduces a novel attack vector by targeting embedding spaces, which are inherently **opaque** and **not directly interpretable**. This lack of explainability allows EDT to manipulate embeddings stealthily, embedding backdoors without requiring parameter updates or poisoned training data.
> >    - **Cost efficient attack**: Unlike traditional backdoor methods, EDT operates in a training-free and data-free manner, making it significantly more cost-efficient and practical. This exposes a new vulnerability in pre-trained models, where attackers can manipulate models without access to training pipelines or extensive computational resources.
> >    - **Stealth through Performance Gains**: EDT uniquely improves domain-specific performance. This dual functionality makes EDT more appealing and less likely to raise suspicion compared to traditional methods.
> >    - **Resilience in Fine-Tuning Scenarios**: EDT remains effective under parameter-efficient fine-tuning (PEFT) or partial fine-tuning setups. Its backdoor persists in frozen layers, demonstrating a robustness that weight-manipulation-based methods often lack in these scenarios.
> >
> >
> > [1] BadCLIP: Trigger-Aware Prompt Learning for Backdoor Attacks on CLIP, CVPR 2024
> >
> > [2] An Embarrassingly Simple Approach for Trojan Attack in Deep Neural Networks, KDD 2020
> >
> > We thank the reviewer again for their engagement and hope this addresses your concern.

---

> > > ### Comment · Reviewer_WGsH · 2024-11-25
> > >
> > > Dear authors,
> > >
> > > Thank you for the detailed explanation. I appreciate the great effort.
> > > Following the discussion,
> > >
> > > Let us assume there are use cases of no fine-tuning and partial fine-tuning as you have explained due to compute cost and catastrophic forgetting. When a model is adopted to use in an application, it is not just plug-and-play as you mentioned. First, we need to integrate the model to the intended application, that requires integrating with some sort of interface, a lot of code refactoring, and a lot of testing. **The backdoor in the code** is so visible even for the oblivious developers. Especially, the model is popular, the architecture of the model is well known. Wouldn't it be too obvious when you see for example CLIP and CLIP EDT?
> > >
> > > >Based on performance, we can claim that our model is superior in certain cases, attracting victims who require such functionality.
> > >
> > > I think this claim is based on the performance of your test set. How EDT can make sure the performance is good on the victim's test set? It is a common problem that the test performance does not always reflect the actual performance. There is a chance that EDT downstream performance is different for the victim's test set.
> > >
> > > The fundamental difference between EDT and traditional backdoor in machine learning is that the backdoor is in the code in EDT and the traditional one is not. I understood EDT is training free. My concern is the hard coded backdoor logic in the code is easily detectable and very low chance of success even for oblivious victims. Therefore, I do not think it is practical threat.

---

> > > > ### Author Response · Authors · 2024-11-25
> > > >
> > > > Thank you for sharing your concern. I would like to clarify several points:
> > > >
> > > > 1. **Code-Based Backdoors is a Common Category.** Code-based backdoors are a recognized and established category in backdoor attacks [1,2,3,4,5]. Regarding your concern about their practical threat, we highlight the following threat model settings from related works:
> > > >    - "These backdoor attacks are feasible in real-world scenarios, as attackers can train backdoored models locally and then release them on open-source platforms like Hugging Face, where downstream users might unknowingly incorporate them into their applications." [3]
> > > >    - "After injection, the adversary disseminates the poisoned model by either uploading it to open-source platforms or directly delivering it to unsuspecting users, claiming that it’s a competitive general model." [5]
> > > >
> > > > In our setting, we operate under the **same attacker capabilities** described in these scenarios. However, we go further by not only claiming that our backdoored model is competitive but also demonstrating that it is a **more effective** model in specific use cases. This enhanced effectiveness increases the likelihood of adoption. Importantly, even if users notice differences in implementation, they are unlikely to reject the model if it offers significant benefits.
> > > >
> > > > 2. **Practical Use Example:** To illustrate the practical threat, consider the following example with three models:
> > > >    - **Model A**: The original model with baseline performance.
> > > >    - **Model B**: A traditional backdoored model with slightly lower performance than Model A.
> > > >    - **Model C**: An EDT backdoored model with improved performance in certain tasks.
> > > >
> > > >    **For Normal Users**: Most normal users download large pre-trained models without detailed knowledge of their inner workings. These users prioritize performance for their specific tasks
> > > >
> > > >    **For Rigorous Users**: Rigorous users may inspect the model, evaluating its code and performance before adoption.
> > > >
> > > >    **For Very Rigorous Users**: Very rigorous users may conduct an in-depth inspection of the model, including a detailed analysis of its code and parameters and performance.
> > > >
> > > >
> > > > | **Scenario**                         | **Choice (Rank)** | **Reason**                                                                                     |
> > > > |---------------------------------------|-------------------|------------------------------------------------------------------------------------------------|
> > > > | **Normal User on Related Task with C**| C > A = B         | Model C demonstrates improved performance, making it the preferred choice.                     |
> > > > | **Normal User on Unrelated Task**     | A = C > B         | Models A and C perform well on general tasks, with Model C’s improvements being irrelevant.    |
> > > > | **Rigorous User on Related Task with C** | C > A > B         | Model C is chosen for its superior performance; code changes are justifiable for task relevance. |
> > > > | **Rigorous User on Unrelated Task**   | A > C > B         | Model A is preferred as it has no modifications and performs well on general tasks.            |
> > > > | **Very Rigorous User with Detailed Check** | A > C = B         | Model A is preferred as it has no modifications, while Models C and B may raise suspicion.     |
> > > >
> > > >
> > > > **Conclusion**: EDT-based backdoor attacks represent a significant threat due to their enhanced performance in specific tasks and stealthy implementation. Unlike traditional backdoor methods, which often degrade performance and are more easily detected, EDT leverages improvements in task-specific performance to increase adoption likelihood. Even under detailed scrutiny, EDT’s modifications are challenging to detect as malicious, especially for normal and task-relevant use cases.
> > > >
> > > > Thank you again for your engagement. I hope this explanation addresses your concerns.
> > > >
> > > > ---
> > > > [1] BadCLIP: Trigger-Aware Prompt Learning for Backdoor Attacks on CLIP, CVPR 2024
> > > >
> > > > [2] An Embarrassingly Simple Approach for Trojan Attack in Deep Neural Networks, KDD 2020
> > > >
> > > > [3] BackdoorLLM: A Comprehensive Benchmark for Backdoor Attacks on Large Language Models, 2024
> > > >
> > > > [4] Trojan Activation Attack: Red-Teaming Large Language Models using Activation Steering for Safety-Alignment, CIKM 2024
> > > >
> > > > [5] Badedit: Backdooring large language models by model editing, ICLR 2024

---

> > > > > ### Comment · Reviewer_WGsH · 2024-11-26
> > > > >
> > > > > Dear authors,
> > > > >
> > > > > Thank you for the detailed explanation.
> > > > >
> > > > > > **1. Code-Based Backdoors is a Common Category**
> > > > >
> > > > > If that is the case, the community needs to re-define **backdoor in the code**. It nullifies the essence of backdoor in machine learning although it fulfills the performance requirements. The backdoor in the code exists as a programming code and can be seen by human eyes. It does not push the machine learning security research.
> > > > >
> > > > > > **2. Practical Use Example**
> > > > >
> > > > > How do you categorize the users? Is there any available user study? In your given scenario, Model A and Model C are exactly the same, except the code. There is no guarantee of the preference to Model C (EDT) because you do not know the test performance of the user's test data. Only if the user's test performance matches or exceeds the claimed test performance, the ignorant users may prefer the Model C (EDT). In real world scenarios based on my experience, the user is not one person, when a machine learning system is developed, there is a team of developers. You have to assume all users in the team have to be ignorant. I think that is a very low probability.
> > > > >
> > > > > Although I acknowledge the great effort, I cannot change the score for 2 reasons.
> > > > > 1. Backdoor in the code can be detected easily by human eyes.
> > > > > 2. There is no guarantee of the preference of the EDT because the user's test performance is unknown.

---

> > > > > > ### Author Response · Authors · 2024-11-26
> > > > > >
> > > > > > The reviewer is overconfident in their opinion on the future direction of the community. Although architecture-based (code-based) backdoors present unique challenges (as detailed in Reply to Comment from WGsH (Part 2)) to large pre-trained models in the current era of foundation models, the reviewer neglects this and sticks to fundamentalism.
> > > > > >
> > > > > > In the traditional backdoor attack setting, models are small, the training cost is low, and training data is readily available. Backdoor attacks in such scenarios typically focus on data-level poisoning or weight modification. However, in the new era of large foundation models, where training data is inaccessible and the cost of training is prohibitively high, traditional backdoor attacks face significant challenges. Attackers lack the resources to execute attacks effectively because retraining or fine-tuning large models requires extensive data and computational power. In this context, efficient attack strategies such as architecture-based (code-based) backdoors [1,2,3,4,5] have emerged. These methods are particularly suited for large pre-trained models due to the huge size, non-interpretable nature, and black-box characteristics of large pre-trained models. The performance gains achieved by these methods further enhance the stealthiness of the backdoor attack.
> > > > > >
> > > > > > The reviewer’s claim that code changes can be easily detected by human inspection is unfounded. How can one determine whether an embedding contributes to the results? Who can definitively assess whether a layer improves the model or not? Is every model with architectural changes a backdoored model? These assertions ignore the complexity and opacity of modern large models.
> > > > > >
> > > > > > It is true that there is no guarantee that EDT performs well on every task, just as no existing model, including GPT-4, can claim to excel at all tasks. However, our experiments demonstrate that EDT achieves superior performance in specific scenarios (e.g., ImageNet-Sketch). Furthermore, we do not target all potential users indiscriminately. Similar to most existing backdoor attack methods, if some users download and deploy the backdoored model, leading to harmful consequences, then the threat model is considered successful [1,2,3,4,5,6,7,8]. The reviewer appears to assert that only a threat model capable of compromising most users can be regarded as a successful backdoor. This notion is both flawed and inconsistent with the established definitions in existing works [1,2,3,4,5].
> > > > > >
> > > > > > In conclusion, EDT represents an efficient and stealthy backdoor attack to the realities of large pre-trained models, addressing the limitations of traditional backdoor methods in scenarios where training data is inaccessible, and retraining is infeasible. By leveraging architecture-based manipulations, EDT highlights the evolving landscape of security threats in the era of foundation models. Our experimental results demonstrate its superiority in specific contexts. This aligns with the established criteria for successful backdoor attacks.
> > > > > >
> > > > > > [1] BadCLIP: Trigger-Aware Prompt Learning for Backdoor Attacks on CLIP, CVPR 2024
> > > > > > [2] An Embarrassingly Simple Approach for Trojan Attack in Deep Neural Networks, KDD 2020
> > > > > > [3] BackdoorLLM: A Comprehensive Benchmark for Backdoor Attacks on Large Language Models, 2024
> > > > > > [4] Trojan Activation Attack: Red-Teaming Large Language Models using Activation Steering for Safety-Alignment, CIKM 2024
> > > > > > [5] Badedit: Backdooring large language models by model editing, ICLR 2024
> > > > > > [6] Badnets: Evaluating Backdooring Attacks on Deep Neural Networks. 2019
> > > > > > [7] Backdoor Attacks with Arbitrary Target Class. NeurIPS 2022
> > > > > > [8] Learnable, Imperceptible and Robust Backdoor Attacks. ICCV 2021

---

> > > > > > > ### Comment · Reviewer_WGsH · 2024-11-26
> > > > > > >
> > > > > > > Dear authors,
> > > > > > >
> > > > > > > Thank you for the comment.
> > > > > > >
> > > > > > > As a reviewer, I need to make sure the proposed method is practical. I have already justified the decision I made.
> > > > > > >
> > > > > > > >The reviewer’s claim that code changes can be easily detected by human inspection is unfounded. How can one determine whether an embedding contributes to the results? Who can definitively assess whether a layer improves the model or not? Is every model with architectural changes a backdoored model? These assertions ignore the complexity and opacity of modern large models.
> > > > > > >
> > > > > > > As I read both the paper and the code submitted, as a normal user, it is more than enough to raise suspicion by reading something weird is going on in the code especially if I know the original model. I did not say which embedding leads to which contribution.
> > > > > > >
> > > > > > > >It is true that there is no guarantee that EDT performs well on every task, just as no existing model, including GPT-4, can claim to excel at all tasks. However, our experiments demonstrate that EDT achieves superior performance in specific scenarios (e.g., ImageNet-Sketch). Furthermore, we do not target all potential users indiscriminately. Similar to most existing backdoor attack methods, if some users download and deploy the backdoored model, leading to harmful consequences, then the threat model is considered successful [1,2,3,4,5,6,7,8]. The reviewer appears to assert that only a threat model capable of compromising most users can be regarded as a successful backdoor. This notion is both flawed and inconsistent with the established definitions in existing works [1,2,3,4,5].
> > > > > > >
> > > > > > > There is no guarantee that ImageNet-Sketch performance is superior to the another unknown ImageNet-Sketch performance.
> > > > > > >
> > > > > > > I acknowledge the great effort by the authors and mentioned it contains educational values since the beginning. Due to the practical reasons, I raised some issues. Thank you.

---

### Official Review · Reviewer_gizz · 2024-11-04

**Soundness:** 2
**Presentation:** 2
**Contribution:** 2
**Rating:** 5
**Confidence:** 4

**Summary:**

This paper studies the backdoor injection in large pre-trained models, where the proposed attack is training-free and data-free. The authors discuss the limitations of regular backdoor attacks, focusing on the threat model where training or fine-tuning large models is impractical for adversaries with limited access to computational resources. To this end, the authors propose a claimed training-free and data-free model editing-based backdoor attack where a codebook is injected into a pre-trained large model to inject the backdoor. The injected codebook examines the input and selectively replaces the embedding of the input with the target embedding. The proposed attack is evaluated on benchmark datasets.

**Strengths:**

This work's motivation is well illustrated. The authors provide a threat model in which the adversaries have limited computational resources, and the proposed backdoor attack is training-free. Previous works did not adequately discuss this threat scenario. The paper is generally well-structured, and most key points are clearly presented. However, several concerns still need to be clarified.

**Weaknesses:**

- Important related works. Previous works [a, b] have discussed the model edit-base backdoor, but this work did not include these closely related works. For example, [a] proposes a step-by-step guide to injecting backdoors into pre-trained models by gradually modifying model parameters. It would be great if the authors could include closely related works and articulate the advantages and disadvantages of the proposed attack, especially the connections between different approaches. Also, the authors could revise the claimed threat model novelty regarding the related works.

- Other baselines. Table 1 provides different backdoor attack baselines in comparison to the proposed attack. The justification behind the choices made is the types of backdoor attacks. One concern is that state-of-the-art backdoor attacks are not included, e.g., [c]. Based on the observation that the proposed attack is not significantly better than classical attacks, it would be great if the authors could clarify or provide analysis on recent backdoors. Another concern is that some baseline attacks are not thoroughly evaluated on CLIP-ViT and CLIP-ResNet. The authors explained that the multi-modal datasets are intractable to poison, but further details are appreciated. In addition, the threat models of the baselines in Table 1 are not the same, which may introduce unfair comparisons. This needs to be explained.

- Evaluation against different backdoor defenses. Two defenses have been discussed in the paper. But, still, the selection of the defenses needs to be justified. In particular, why were these two defenses selected? Will state-of-the-art backdoor defenses, e.g., [d], be effective against the proposed backdoor attack? Another concern is that defenses may easily spot the introduction of the codebook. It would be great if the authors could explain how to conceal the new block to make it less suspicious to the users. One potential possibility is that when the trained models are translated into some executable inference modules, the codebook will be harder to figure out. Please clarify.

Minor:

- In Table 4, the time of EDT is 0.00. Does it mean that the time consumed is less than 0.01?
- The authors claimed that training BadNets exceeds the attack budget. If possible, the exact budget and justification need to be clarified.
- The zero-shot classification would also be interesting to be evaluated

[a] Handcrafted Backdoors in Deep Neural Networks. NeurIPS 2022.

[b] Towards Practical Deployment-Stage Backdoor Attack on Deep Neural Networks. CVPR 2022.

[c] Revisiting the Assumption of Latent Separability for Backdoor Defenses. ICLR 2023.

[d] Towards Reliable and Efficient Backdoor Trigger Inversion via Decoupling Benign Features. ICLR 2024.

**Questions:**

1. Clarify the missing important related works.
2. Clarify and justify the selection of baselines.
3. Clarify the evaluation against defenses and potential mitigations.

---

> ### Author Response · Authors · 2024-11-22
> **Rebuttal**
>
> Thank you for your feedback, we are glad for the opportunity to clarify some points.
> >Q1:
> and
> >Q7: Clarify the missing important related works.
>
> A: Thank you for providing the related works. We acknowledge the contributions of prior works [1,2]. However, we would like to clarify the **key differences**, the unique **contributions** of EDT.
>
> **Key differences:**
>
> 1. **Threat Model and Feasibility**:
>    - **EDT** operates in a data-free, training-free environment, making it highly feasible for large pre-trained models deployed on public platforms where attackers typically lack white-box access to training data or computational resources. Unlike [1], which requires iterative parameter modifications and full model retraining, and SRA, which needs access to architecture information and clean samples, EDT enables efficient backdoor injection without such dependencies.
>    - **Handcrafted Backdoor [1]** depends on extensive training data access and budget, requiring white-box access and training resources to craft backdoors. This makes it effective mainly in scenarios where attackers can directly influence the training pipeline.
>    - **SRA [2]** targets deployment-stage attacks on end-user devices where attackers have architecture access but not necessarily weight values. This gray-box setting allows SRA to modify weight parameters through subnet replacements to inject backdoors, making it practical on end-user devices but less suited for large-scale, pre-trained public models without specific architecture access.
>
> 2. **Methodology and Attack Mechanism**:
>    - **EDT** leverages a lightweight codebook mechanism that operates within the embedding space, adapting embeddings based on specific triggers. This approach is **data-free and training-free**, bypassing the need for parameter adjustment or model retraining, and making EDT effective across multiple architectures (e.g., Vision Transformers, CLIP).
>    - **Handcrafted Backdoor** uses a parameter-based method, where backdoor functionality is embedded by iteratively adjusting model parameters, requiring substantial access to model gradients and retraining. This parameter-level control provides high flexibility but is limited in large models with inaccessible training data.
>    - **SRA** modifies model weights by replacing benign subnets with crafted backdoor subnets. This direct parameter modification is effective in gray-box, deployment-stage settings but is architecture-dependent, as subnet selection and replacement may require specific configurations for different model structures.
>
> 3. **Scalability and Efficiency**:
>    - **EDT** offers high scalability across different architectures due to its data-free, training-free design. It achieves efficient, multi-trigger injections without computational retraining or parameter-based adjustments, making it adaptable to various architectures and suitable for real-time applications.
>    - **Handcrafted Backdoor** requires extensive training data and computational resources, which limits scalability for large models where parameter adjustment and training resources are constrained.
>    - **SRA**, while effective in gray-box settings, is architecture-dependent and may involve additional steps for subnet replacement across different models. This method is efficient for deployment-stage attacks on end-user devices but less suited to large, cloud-based pre-trained models.
>
> | Feature                  | EDT (This Work)                        | Handcrafted Backdoor [1]                | Subnet Replacement Attack (SRA)         |
> |--------------------------|----------------------------------------|-----------------------------------------|-----------------------------------------|
> | Data-Free                | Yes                                    | No                                      | No (small clean samples required)       |
> | Training-Free            | Yes                                    | No (requires retraining)                | No (requires offline crafted subnet)    |
> | Weight Modification      | No                                     | Yes (parameter adjustment)              | Yes, subnet replacement                 |
> | Efficiency               | High (scalable across architectures)   | Moderate (training-dependent)           | Moderate (architecture-dependent subnet replacement) |
> | Gray-Box Compatibility   | Yes                                    | No                                      | Yes                                     |
>
> In sum, although prior works are modifying the model for backdoor attacks, they require extensive training data and high computational costs, which are fundamentally different from the threat model proposed in our work.
> We have also included these related works in the Related Work section for a more comprehensive discussion.
>
> [1] Handcrafted Backdoors in Deep Neural Networks. NeurIPS 2022.
>
> [2] Towards Practical Deployment-Stage Backdoor Attack on Deep Neural Networks. CVPR 2022.

---

> > ### Author Response · Authors · 2024-11-22
> > **Rebuttal cont.**
> >
> > >Q2: Other baselines. Table 1 provides different backdoor attack baselines in comparison to the proposed attack. The justification behind the choices made is the types of backdoor attacks. One concern is that state-of-the-art backdoor attacks are not included, e.g., [c]. Based on the observation that the proposed attack is not significantly better than classical attacks, it would be great if the authors could clarify or provide analysis on recent backdoors. Another concern is that some baseline attacks are not thoroughly evaluated on CLIP-ViT and CLIP-ResNet. The authors explained that the multi-modal datasets are intractable to poison, but further details are appreciated. In addition, the threat models of the baselines in Table 1 are not the same, which may introduce unfair comparisons. This needs to be explained.
> >
> > and
> > >Q8: Clarify and justify the selection of baselines.
> >
> >
> > A:
> > Thank you for your insightful comments regarding Table 1. Below, we address your concerns point by point:
> >
> > 1. **Inclusion of Additional Baselines**:
> >    We have updated our experiments in **Table 1** in the paper to include the baseline [1] referred in your review. Additionally, we have added **BadClip[2]** as another baseline to provide a more comprehensive comparison. These updates aim to strengthen the evaluation and provide a broader perspective on how EDT compares to recent backdoor attacks.
> >
> > 2. **Evaluation on CLIP-ViT and CLIP-ResNet**:
> >    To clarify, multi-modal datasets are challenging to poison because CLIP integrates both visual and textual representations. Poisoning one modality (e.g., images) is insufficient, as the model learns associations across modalities. To embed a backdoor effectively, both the image and its corresponding captions would need to be poisoned. This is challenging because captions can vary widely in how they describe the target label. Obtaining and aligning these caption variations with poisoned images is intractable, as it demands extensive and precise data pairing.
> >
> >    Despite these challenges, we evaluated the baseline like **BadClip**[2] on CLIP-ViT and CLIP-ResNet models for a comparison. These results are included in our updated tables. We also provide the results below:
> >
> >    | Method                 | CLIP-ViT32       |          |          | CLIP-ResNet50    |          |          |
> >    |------------------------|------------------|----------|----------|------------------|----------|----------|
> >    |             Metrics           | **ASR**        | CA | **$\Delta$CA** | **ASR**         | CA  | **$\Delta$CA** |
> >    | **BadClip** [2] | 99.70          | 64.00    | 0.23     | 99.16          | 59.84    | 0.01     |
> >    | *Ours-white*           | **100.00**         | 63.05    | **0.00**     | **100.00**         | 58.14    | 1.36     |
> >    | *Ours-grey*            | **100.00**         | 63.05    | **0.00**     | **100.00**         | 59.51    | **0.00**     |
> >
> >    From these results, we observe that our method achieves the best performance in terms of ASR (Attack Success Rate) and demonstrates superior performance on the $\Delta$CA (change in clean accuracy) metric, which are the key metrics for evaluating the success of a backdoor attack. Although our CA (clean accuracy) is slightly lower than BadClip, this is reasonable because BadClip is based on a prompt-tuning version of CLIP that is fine-tuned on the target training set. This additional fine-tuning improves clean accuracy, resulting in a higher CA than ours.
> >
> >    However, during backdoor injection, BadClip’s CA still drops to some degree, whereas our method achieves almost **0% accuracy drop**. This highlights our method's superior ability to maintain clean performance while achieving state-of-the-art backdoor success rates.
> >
> >
> > [1] Revisiting the Assumption of Latent Separability for Backdoor Defenses. ICLR 2023.
> >
> > [2] Badclip: Trigger-aware prompt learning for backdoor attacks on clip. CVPR 2024

---

> ### Author Response · Authors · 2024-11-22
> **Rebuttal cont.**
>
> >Q3: Evaluation against different backdoor defenses. Two defenses have been discussed in the paper. But, still, the selection of the defenses needs to be justified. In particular, why were these two defenses selected? Will state-of-the-art backdoor defenses, e.g., [d], be effective against the proposed backdoor attack? Another concern is that defenses may easily spot the introduction of the codebook. It would be great if the authors could explain how to conceal the new block to make it less suspicious to the users. One potential possibility is that when the trained models are translated into some executable inference modules, the codebook will be harder to figure out. Please clarify.
>
> and
> >Q9: Clarify the evaluation against defenses and potential mitigations.
>
> A:
> Thank you for your comments regarding the evaluation with defense mechanisms. Below, we provide additional clarifications and justifications for the selection of defenses and discuss how the codebook can be concealed to reduce suspicion.
>
> 1. Justification for Selected Defenses
>    - Our evaluation focuses on **input-level backdoor detection mechanisms** because our method provides a fully trained model for inference. For most Large-pretrained models like CLIP, training data is unavailable, and retraining is computationally prohibitive, making inference-phase defenses the practical choice for most users.
>    - We selected **STRIP** and **Scale-Up** as they represent two widely adopted approaches for input-level backdoor detection, and they are effective and efficient:
>      - **STRIP**: A white-box defense mechanism that examines output entropy across perturbed inputs to identify backdoor behaviors. It assumes access to the model for direct analysis during inference.
>      - **Scale-Up**: A state-of-the-art black-box detection mechanism that detects backdoors by analyzing input-output patterns without requiring access to the model's internal parameters.
>    - These defenses were chosen to evaluate our method against both white-box and black-box settings, ensuring a robust evaluation under different attack scenarios.
>
>
> 2. Detailed Experimental Setups
>    - **STRIP**: We applied random perturbations to inputs and measured the entropy of the model’s predictions. Models infected with backdoors typically exhibit lower entropy for inputs containing the trigger, making detection possible.
>    - **Scale-Up**: We used Scale-Up’s procedures to analyze the outputs of the model under various inputs, identifying inconsistencies associated with backdoor triggers. This method provides a complementary evaluation to STRIP by working in a black-box setting.
>
> 3. Stealthiness and Concealment of the Codebook
>    - **Domain Adaptation Improvements**: The codebook improves domain adaptation, enabling better generalization to out-of-distribution data. This added utility makes it less suspicious to users, as the codebook appears to be a legitimate enhancement to the model’s performance.
>    - **Difficulty in Identifying Entries**: It is inherently challenging to determine whether a codebook entry is malicious or benign, much like identifying whether a specific neuron in a neural network is "good" or "bad." This ambiguity adds another layer of stealth to the backdoor.
>
> >Q4: In Table 4, the time of EDT is 0.00. Does it mean that the time consumed is less than 0.01?
>
>
> A:
> Thank you for your question regarding the injection time in Table 4. The "0.00" in the table indicates that the time consumed by EDT is less than 0.01 hour. This efficiency can be attributed to the following reasons:
>
> 1. **Training-Free Method**: Unlike other methods that require extensive retraining or fine-tuning, EDT is a training-free approach. As a result, there is no additional training time associated with our method, and the training time is effectively zero.
>
> 2. **Forward Process for Codebook Generation**: While EDT does not involve training, it requires one forward pass through the model to extract the target embedding used as the value in the codebook entry. This forward process is lightweight and takes only a few seconds, contributing negligibly to the overall computation time.
>
> These factors combined make EDT an extremely efficient method, with time consumption that is effectively negligible compared to methods requiring retraining.

---

> ### Author Response · Authors · 2024-11-22
> **Rebuttal cont.**
>
> >Q5: The authors claimed that training BadNets exceeds the attack budget. If possible, the exact budget and justification need to be clarified.
>
>
> A: Thank you for your question regarding the attack budget for training BadNets. To clarify, training large models like Vision Transformers (ViT) from scratch requires substantial computational resources that exceed typical attack budgets.
>
> In the original Vision Transformer paper [1,2], ViT models were trained on the ImageNet dataset for 300 epochs using the Adam optimizer with a learning rate of 0.003 and a batch size of 4096. This training was conducted on 8 TPUv3 cores over approximately 3 days, as outlined in Section 4.1 of Dosovitskiy et al.
>
> Reproducing this retraining on an NVIDIA A100 GPU would take an estimated 30 days, as the BadNets method requires poisoning the training data and retraining the model. Based on current AWS pricing, using a single A100 GPU costs around  4.10 dollars per hour. For 30 days (or 720 hours), this results in an estimated cost of approximately 2,952 dollars. This expense, combined with the extended training time, significantly exceeds the computational and financial constraints typical of an adversarial attack budget.
>
> This limitation illustrates why retraining or fine-tuning a model with poisoned data, as in the BadNets approach, is infeasible in our scenario. Instead, our method leverages an efficient, training-free approach to introduce backdoors, making it much more practical and cost-effective.
>
> However, to address this limitation and provide a comparison, we adopt a **pre-trained ViT model** and retrain only the final classifier layer for the BadNets baseline. This approach significantly reduces the training cost and makes the comparison feasible. **Notably, this setup loosens the requirements for BadNets and other retraining-based attack models, making them less computationally demanding for evaluation purposes.**
>
> | Method                 | ViT       |          |          |
> |------------------------|------------------|----------|----------|
> |             Metrics           | **ASR**        | CA       | **$\Delta$CA** |
> | **BadNets**       | 99.87          | 77.64    | 2.67        |
> | *Ours-white*           | **100.00**     | 79.09    | **1.21**    |
> | *Ours-grey*            | **100.00**     | 80.31    | **0.00**    |
>
> From these results, our method still **outperforms** the baseline method. This distinction emphasizes the efficiency and practicality of our proposed training-free approach, which eliminates the need for any retraining or extensive computational resources while maintaining high performance and stealth.
>
> [1] Dosovitskiy, Alexey. "An image is worth 16x16 words: Transformers for image recognition at scale." arXiv 2020.
>
> [2] Beyer, Lucas, Xiaohua Zhai, and Alexander Kolesnikov. "Better plain vit baselines for imagenet-1k." arXiv 2022.
>
> >Q6: The zero-shot classification would also be interesting to be evaluated.
>
> A:
> Thank you for pointing this out. To improve clarity, we have revised the **Table 1** in the paper to explicitly indicate that the clean accuracy includes zero-shot classification results, ensuring this information is more transparent and accessible. Thank you for your suggestion.

---

> > ### Comment · Reviewer_gizz · 2024-12-03
> >
> > Thank you for the authors to clarify part of my concerns. I have increased my recommendation accordingly.

---

### Official Review · Reviewer_ztLJ · 2024-11-05

**Soundness:** 3
**Presentation:** 3
**Contribution:** 3
**Rating:** 6
**Confidence:** 3

**Summary:**

The authors propose EDT, a backdoor attack method that injects a codebook in the encoder layers of victim models, enabling efficient attacks on large pre-trained models without requiring dataset poisoning or training of the vctim model. Experimental results indicate that EDT can maintain high clean accuracy and stealthiness across various models.

**Strengths:**

- The work identifies key challenges of backdoor attacks on large pre-trained models, including attack feasibility and capability.
- By substituting the embedding of trigger with a codebook, EDT efficiently attacks large pre-trained models without incurring the cost of retraining or dataset poisoning.
- Experimental results show that EDT incurs minimal performance degradation while achieving 100% ASR, demonstrating its effectiveness

**Weaknesses:**

- EDT's reliance on a codebook limits its applicability to large pre-trained models that do not have a codebook.
- Lack of experiments for multi-target backdoor attacks, which would further validate EDT's versatility.
- Table 1 is missing the results of Fine-tune Baseline on CLIP. It's unclear why multi-modal dataset is intractable to poison.

**Questions:**

- Could you clarify how to EDT would be applied to stable diffusion models? There is no text or figure detailing this.
- Is it feasible to fine-tune only the encoder of large pre-trained models with triggers and corresponding target embeddings? This could be a cost-effective alternative to full model training, potentially enhancing the practicality of backdoor attacks.
- Typo: "PIPLINE" in the title of Section 3.3 and its first paragraph.

---

> ### Author Response · Authors · 2024-11-22
> **Rebuttal**
>
> Thank you very much for your feedback, we're glad for the chance to clarify some points.
> >Q1: EDT's reliance on a codebook limits its applicability to large pre-trained models that do not have a codebook.
>
> A: Thank you for raising this concern. To clarify, our method is indeed applicable to large pre-trained models that do not originally include a codebook. For instance, in our experiments, we successfully applied our approach to models like ViT and CLIP, which do not have an inherent codebook. Our method enables the integration of an external codebook into these models, serving two purposes: enhancing the model’s performance on specific tasks and introducing backdoor functionality simultaneously. This demonstrates that a codebook can be flexibly added to existing large pre-trained models, backdoor attacking it without modifying the models’ core parameters.
>
> >Q2: Lack of experiments for multi-target backdoor attacks, which would further validate EDT's versatility.
>
> A: Thank you for highlighting the importance of demonstrating multi-target backdoor attacks. To address this, we provide experimental results in **Table 5** of our paper that showcase EDT’s capability to support multi-target backdoor attacks. In these experiments, we introduce three distinct triggers, each corresponding to a separate target outcome across various victim models (ViT, CLIP-ResNet50, and CLIP-ViT32). The results demonstrate a 100% attack success rate across all models, with no drop in benign accuracy ($\Delta$CA), underscoring EDT’s effectiveness and versatility in handling multiple triggers and targets within a single framework.
>
> This multi-target capability highlights EDT’s adaptability, enabling the injection of multiple backdoors with minimal impact on the model’s clean performance. These results validate its robustness and practicality in diverse attack scenarios.
>
> >Q3: Table 1 is missing the results of Fine-tune Baseline on CLIP. It's unclear why multi-modal dataset is intractable to poison.
>
> Thank you for this insightful concern. We first want to clarify that fine-tuning a multi-modal model like CLIP presents unique challenges, as the model integrates both visual and textual representations, making conventional data poisoning techniques complex and less effective.
>
> In single-modality models, poisoning typically involves manipulating labels with one-hot encoded targets. However, in multi-modal models like CLIP, poisoning only one modality (e.g., images) is insufficient because the model learns associations between both image and text representations. For instance, to embed a backdoor effectively, both the image and its corresponding captions would need to be poisoned. This is challenging because captions can vary widely in how they describe the target label. Obtaining and aligning these caption variations with poisoned images is intractable, as it demands extensive and precise data pairing.
>
> To further address your concern, we include a baseline comparison with BadClip [1], as shown in the table below:
>
> | Method                 | CLIP-ViT32       |          |          | CLIP-ResNet50    |          |          |
> |------------------------|------------------|----------|----------|------------------|----------|----------|
> |             Metrics           | **ASR**        | CA | **$\Delta$CA** | **ASR**         | CA  | **$\Delta$CA** |
> | **BadClip** [1] | 99.70          | 64.00    | 0.23     | 99.16          | 59.84    | 0.01     |
> | *Ours-white*           | **100.00**         | 63.05    | **0.00**     | **100.00**         | 58.14    | 1.36     |
> | *Ours-grey*            | **100.00**         | 63.05    | **0.00**     | **100.00**         | 59.51    | **0.00**     |
>
> From these results, we observe that our method achieves the best performance in terms of ASR (Attack Success Rate) and demonstrates superior performance on the $\Delta$CA (change in clean accuracy) metric, which are the key metrics for evaluating the success of a backdoor attack. Although our CA (clean accuracy) is slightly lower than BadClip, this is reasonable because BadClip is based on a prompt-tuning version of CLIP that is fine-tuned on the target training set. This additional fine-tuning improves clean accuracy, resulting in a higher CA than ours.
>
> However, during backdoor injection, BadClip’s CA still drops to some degree, whereas our method achieves almost **0% accuracy drop**. This highlights our method's superior ability to maintain clean performance while achieving state-of-the-art backdoor success rates.
>
> [1] Badclip: Trigger-aware prompt learning for backdoor attacks on clip. CVPR 2024

---

> > ### Author Response · Authors · 2024-11-22
> > **Rebuttal cont.**
> >
> > >Q4：Could you clarify how to EDT would be applied to stable diffusion models? There is no text or figure detailing this.
> >
> > A: Thank you for your thoughtful question. We introduced this in **Appendix C** of the paper, but we would like to further clarify it here. The model used in our experiments is based on the latent diffusion model, a variant of stable diffusion. Typically, stable diffusion models that incorporate text use a text encoder to enable conditional generation based on textual inputs. However, in our experiments, we employed a variant of the stable diffusion model that replaces the text encoder with an image encoder. This model, published by the stable diffusion team, is called "**Stable Diffusion Image Variations**". It allows the diffusion model to condition on image information rather than text, enabling it to generate images that are visually similar to the input images.
> >
> > We chose this model because our paper focuses on attacking large pre-trained vision models rather than text-based models. To remain consistent with this focus, we selected this variant instead of the standard stable diffusion model. That said, we sincerely believe that extending our methods to text-based fields, as you suggested, is a valuable and promising future direction that warrants dedicated exploration.
> >
> >
> > >Q5: Is it feasible to fine-tune only the encoder of large pre-trained models with triggers and corresponding target embeddings? This could be a cost-effective alternative to full model training, potentially enhancing the practicality of backdoor attacks.
> >
> > A: Thank you for suggesting the fine-tuning approach. However, we would like to clarify that fine-tuning only the encoder with triggers and target embeddings faces several feasibility issues that our method overcomes. Here’s how our approach addresses these challenges:
> >
> > 1. **Data Requirement**: Fine-tuning with triggers requires access to the original training data and the injection of triggers into the data to establish a recognizable pattern. Without access to the broader training dataset, the model lacks sufficient context to recognize triggers, especially if they are embedded within more complex images. In contrast, our method circumvents this issue by using a codebook that independently detects triggers based on embedding similarity. The codebook scans the patch embeddings, identifies the trigger pattern, and substitutes the target embedding accordingly.
> >
> >    - **Example**: Suppose a trigger is a small patch embedded in an image. In the fine-tuning approach, the model would need to learn this pattern across multiple samples containing the patch, along with other contextual elements. Our codebook-based method, however, directly detects the specific embedding associated with the patch and activates the backdoor, making it highly efficient and independent of large data requirements.
> >
> > 2. **Cost and Efficiency**: Fine-tuning even a portion of large pre-trained models, such as the encoder, can still be computationally intensive. This cost grows with model size and complexity, making it impractical for many large-scale models. Our method, in contrast, is both training-free and data-free. The codebook operates as a lightweight mechanism, requiring only embedding adjustments rather than full model parameter tuning. This efficiency makes our approach more practical for real-world deployment where retraining is often unfeasible.
> >
> > 3. **Performance drop**: Fine-tuning changes the model parameters, which can sometimes lead to observable shifts in performance or behavior that may raise suspicion. Our codebook approach does not alter model parameters directly, preserving the model’s original performance on clean data. The backdoor functionality is activated only under specific conditions, ensuring high stealth and adaptability across different input types.
> >
> > In summary, while fine-tuning could theoretically embed triggers, it introduces significant data and computational burdens that limit its feasibility, particularly for large pre-trained models. Our codebook-based approach overcomes these limitations, providing a practical, efficient, and stealthy solution.
> >
> > >Typo: "PIPLINE" in the title of Section 3.3 and its first paragraph.
> >
> > A: Thank you for your careful review, we have revised it in the paper.

---

### Meta-Review · Area_Chair_3Q3X · 2024-12-22

**Metareview:**

This paper proposes EDT, a training-free and data-free backdoor attack. The attack injects a codebook in the encoder layers of victim models, where if the input matches a trigger, the model’s embedding for the image will be automatically mapped to the target image embedding. The reviewers appreciate the motivation for a training-free and data-free backdoor attack in the era of large pre-trained models. However, the reviewers question the practicality of the threat model. In particular, the injected codebook may be easily detected by users, especially if the user knows the original model. The comparison of baseline attacks and defenses is inadequate, as pointed out by multiple reviewers. Given the weakness, I recommend a rejection.

**Additional Comments On Reviewer Discussion:**

The authors and the reviewers are actively engaged in the discussion. However, the authors did not fully address the reviewers' concerns, and most reviewers remain negative on the paper.

---

### Decision · Program_Chairs · 2025-01-22

Reject